# Are Large-scale Soft Labels Necessary for Large-scale Dataset Distillation?

**Lingao Xiao[1,3]** and **Yang He[1,2,3]** *

[1]CFAR, Agency for Science, Technology and Research, Singapore
[2]IHPC, Agency for Science, Technology and Research, Singapore
[3]National University of Singapore
xiao_lingao@u.nus.edu, he_yang@cfar.a-star.edu.sg

## Abstract

In ImageNet-condensation, the storage for auxiliary soft labels exceeds that of the condensed dataset by over 30 times. However, *are large-scale soft labels necessary for large-scale dataset distillation*? In this paper, we first discover that the high within-class similarity in condensed datasets necessitates the use of large-scale soft labels. This high within-class similarity can be attributed to the fact that previous methods use samples from different classes to construct a single batch for batch normalization (BN) matching. To reduce the within-class similarity, we introduce class-wise supervision during the image synthesizing process by batching the samples within classes, instead of across classes. As a result, we can increase within-class diversity and reduce the size of required soft labels. A key benefit of improved image diversity is that soft label compression can be achieved through simple random pruning, eliminating the need for complex rule-based strategies. Experiments validate our discoveries. For example, when condensing ImageNet-1K to 200 images per class, our approach compresses the required soft labels from 113 GB to 2.8 GB ($40\times$ compression) with a 2.6% performance gain. Code is available at: https://github.com/he-y/soft-label-pruning-for-dataset-distillation.

## 1 Introduction

We are pacing into the era of ImageNet-level condensation, and the previous works [1, 2, 3, 4, 5] fail in scaling up to large-scale datasets due to extensive memory constraint. Until recently, Yin *et al.*[6] decouple the traditional distillation scheme into three phases. First, a teacher model is pretrained with full datasets (squeeze phase). Second, images are synthesized by matching the Batch Normalization (BN) statistics from the teacher and student models (recover phase). Third, auxiliary data such as soft labels are pre-generated from different image augmentations to create abundant supervision for post-training (relabel phase).

However, the auxiliary data are **30×** larger than the distilled data in ImageNet-1K. To

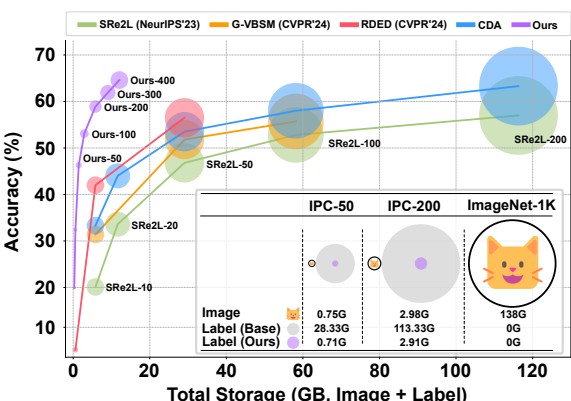

Figure 1: The relationship between performance and total storage of auxiliary information needed. Our method achieves SOTA performance with **fewer soft labels** than images.

---

*Corresponding Author

38th Conference on Neural Information Processing Systems (NeurIPS 2024).

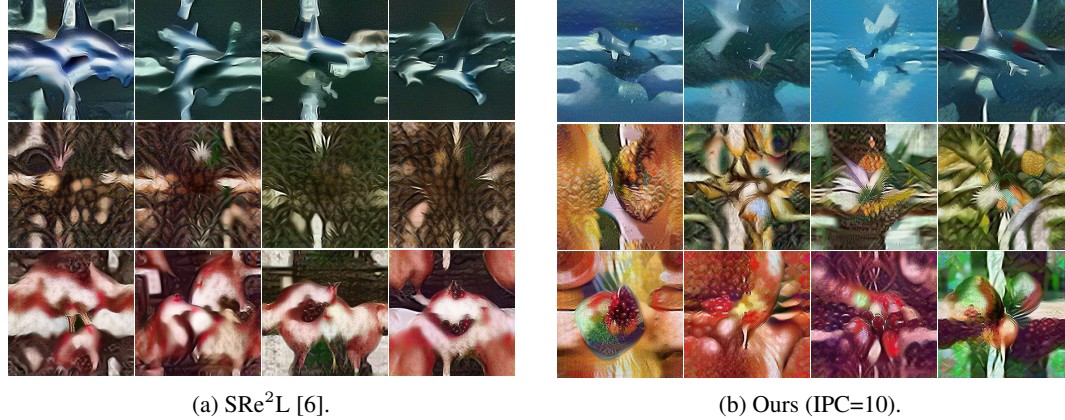

(a) SRe$^2$L [6].  (b) Ours (IPC=10).

Figure 2: Visual comparison between SRe$^2$L and the proposed method. The classes are hammer shark (top), pineapple (middle), and pomegranate (bottom). Our method is more visually diverse.

attain correct and effective supervision, the exact augmentations and soft labels of every training epoch are stored [6, 7, 8, 9, 10]. The required soft label storage is the colored circles in Fig. 1.

In this paper, we consider *whether large-scale soft labels are necessary*, and *what causes the excessive requirement of these labels*? To answer these questions, we provide an analysis of the distilled images using SRe$^2$L [7], and we find that within-class diversity is at stake as shown in Fig. 2. To be more precise, we analyze the similarity using Feature Cosine Similarity and Maximum Mean Discrepancy in Sec. 3.2. The high similarity of images within the same class requires extensive data augmentation to provide different supervision.

To address this issue, we propose **Label Pruning for Large-scale Distillation (LPLD)**. Specifically, we modified the algorithms by batching images within the same class, leveraging the fact that different classes are naturally independent. Furthermore, we introduce class-wise supervision to align our changes. In addition, we have explored different label pruning metrics and found that simple random pruning was performed on par with carefully selected labels. To further increase diversity, we improve the label pool by introducing randomness in a finer granularity (i.e., batch-level). Our method effectively distills the images while requiring less label storage compared to image storage, as shown in Fig. 1.

The key contributions of this work are: (1) To the best of our knowledge, it is the first work to introduce label pruning to large-scale dataset distillation. (2) We discover that high within-class diversity necessitates large-scale soft labels. (3) We re-batch images and introduce class-wise supervision to improve data diversity, allowing random label pruning to be effective with an improved label pool. (4) Our LPLD method achieves SOTA performance using a lot less label storage, and it is validated with extensive experiments on various networks (e.g., ResNet, EfficientNet, MobileNet, and Swin-V2) and datasets (e.g., Tiny-ImageNet, ImageNet-1K, and ImageNet-21K).

## 2 Related Works

**Dataset Distillation.** DD [1] first introduces dataset distillation, which aims to learn a synthetic dataset that is equally effective but much smaller in size. The matching objectives include performance matching [1, 11, 12, 13, 14], gradient matching [4, 15, 16, 17], distribution or feature matching [5, 2, 18], trajectory matching [3, 19, 20], representative matching [21, 22], loss-curvature matching [23], and Batch-Norm matching[6, 7, 9, 10].

**Dataset Distillation of Large-Scale Datasets.** Large-scale datasets scale up in terms of image size and the number of total images, incurring affordable memory consumption for most of the well-designed matching objectives targeted for small datasets. MTT [3] is able to condense Tiny-ImageNet (ImageNet-1K subsets with images downsampled to $64 \times 64$ and 200 classes). IDC [24] conducts experiments on ImageNet-10, which contains an image size of $224 \times 224$ but has only 10 classes. TESLA [20] manages to condense the full ImageNet-1K dataset by exactly computing the unrolled

gradient with constant memory or complexity. SRe$^2$L [6] decouples the bilevel optimization into three phases: 1) squeezing, 2) recovering, and 3) relabeling. The proposed framework surpasses TESLA [20] by a noticeable margin. CDA [7] improves the recovering phase by introducing curriculum learning. RDED [8] replaces the recovering phase with an optimization-free approach by concatenating selected image patches. SC-DD [10] uses self-supervised models as recovery models. Existing methods [7, 8, 10] place high emphasis on improving the recovering phase; however, the problem of the relabeling phase is overlooked: *a large amount of storage is required for the relabeling phase.*

**Label Compression.** The problem of excessive storage seems to be fixed if the teacher model generates soft labels immediately used by the student model on the fly. However, when considering the actual use case of distilled datasets (i.e., Neural Architecture Search), using pre-generated labels enjoys speeding up training and reduced memory cost. More importantly, the generated labels can be repeatedly used. FKD [25] employs label quantization to store only the top-$k$ logits. In contrast, our method retains full logits, offering an orthogonal approach to quantization. A comparison to FKD is provided in Appendix D.3. Unlike FerKD [26], which removes some unreliable soft labels, our strategy targets higher pruning ratios.

**Comparison with G-VBSM [9].** In one recent work, G-VBSM also mentioned re-batching the images within classes; however, the motivation is that having a single image in a class is insufficient [9]. It re-designed the loss by introducing a model pool, matching additional statistics from convolutional layers, and updating the statistics of synthetic images using exponential moving averages (EMA). Additionally, an ensemble of models is involved in both the data synthesis and relabel phase, requiring a total of $N$ forward propagation from $N$ different models, where $N = 4$ is used for ImageNet-1K experiments. On the other hand, we aim to improve the within-class data diversity for **reducing soft label storage**. Furthermore, to account for the re-batching operation, we introduce class-wise supervision while all G-VBSM statistics remain global.

## 3 Method

### 3.1 Preliminaries

The conventional Batch Normalization (BN) transformation is defined as follows:

$$y = \gamma \left( \frac{\boldsymbol{x} - \mu}{\sqrt{\sigma^2 + \epsilon}} \right) + \beta, \tag{1}$$

where $\gamma$ and $\beta$ are parameters learned during training, $\mu$ and $\sigma^2$ are the mean and variance of the input features, and $\epsilon$ is a small constant to prevent division by zero. Additionally, the running mean and running variance are maintained during network training and subsequently utilized as $\mu$ (mean) and $\sigma^2$ (variance) during the inference phase, given that the true mean and variance of the test data are not available.

The matching object of SRe2L [7] follows DeepInversion [27], which optimizes synthetic datasets by matching the models' layer-wise BN statistics:

$$
\begin{aligned}
\mathcal{L}_{\mathrm{BN}}(\widetilde{\boldsymbol{x}}) &= \sum_l \left\| \mu_l(\widetilde{\boldsymbol{x}}) - \mathbb{E}\left( \mu_l \mid \mathcal{T} \right) \right\|_2 + \sum_l \left\| \sigma_l^2(\widetilde{\boldsymbol{x}}) - \mathbb{E}\left( \sigma_l^2 \mid \mathcal{T} \right) \right\|_2 \\
&\approx \sum_l \left\| \mu_l(\widetilde{\boldsymbol{x}}) - \mathbf{BN}_l^{\mathrm{RM}} \right\|_2 + \sum_l \left\| \sigma_l^2(\widetilde{\boldsymbol{x}}) - \mathbf{BN}_l^{\mathrm{RV}} \right\|_2,
\end{aligned} \tag{2}
$$

where the BN's running mean $\mathbf{BN}_l^{\mathrm{RM}}$ and running variance $\mathbf{BN}_l^{\mathrm{RV}}$ are used to approximate the expected mean $\mathbb{E}\left( \mu_l \mid \mathcal{T} \right)$ and expected variance $\mathbb{E}\left( \sigma_l^2 \mid \mathcal{T} \right)$ of the original dataset $\mathcal{T}$, repsectively. The BN loss matches BN for layers $l$, and $\mu_l(\widetilde{\boldsymbol{x}})$ and $\sigma_l^2(\widetilde{\boldsymbol{x}})$ are the mean and variance of the synthetic images $\widetilde{\boldsymbol{x}}$.

The BN loss term is used as a regularization term applied to the classification loss $\mathcal{L}_{\mathrm{CE}}$. Therefore, the matching objective is:

$$\arg\min_{\widetilde{\boldsymbol{x}}} \underbrace{\ell\left( \boldsymbol{\theta}_{\mathcal{T}}\left( \widetilde{\boldsymbol{x}} \right), \boldsymbol{y} \right)}_{\mathcal{L}_{\mathrm{CE}}} + \alpha \cdot \mathcal{L}_{\mathrm{BN}}\left( \widetilde{\boldsymbol{x}} \right), \tag{3}$$

where $\boldsymbol{\theta}_{\mathcal{T}}$ is the model pretrained on the original dataset $\mathcal{T}$. The symbol $\alpha$ is a small factor controlling the regularization strength of BN loss.

Table 1: The cosine similarity between image features. The similarities are the average of 1K class on the synthetic ImageNet-1K dataset. Features are extracted using pretrained ResNet-18.

| IPC | SRe$^2$L | CDA | Ours | Full Dataset |
|-----|---------|-----|------|--------------|
| 50 | $0.841 \pm 0.023$ | $0.816 \pm 0.026$ | $0.796 \pm 0.029$ | |
| 100 | $0.840 \pm 0.016$ | $0.814 \pm 0.019$ | $0.794 \pm 0.021$ | $0.695 \pm 0.045$ |
| 200 | $0.839 \pm 0.011$ | $0.813 \pm 0.013$ | $0.793 \pm 0.015$ | |

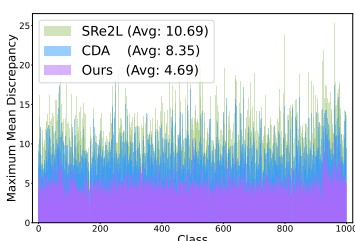

Figure 3: MMD visualization.

## 3.2 Diversity Analysis on Synthetic Dataset

### 3.2.1 Similarity within Synthetic Dataset: Feature Cosine Similarity

A critical aspect of image diversity is how similar or different the images are within the same class. To quantify this, we utilize the feature cosine similarity measure defined above. Lower cosine similarity values between images within the same class indicate greater diversity, as the images are less similar to one another. This relationship is formally stated as follows:

**Proposition 1.** *The lower feature cosine similarity of images indicates higher diversity because the images are less similar to one another.*

Feature Cosine similarity can be formally put as:

$$\cos \text{similarity} := \frac{f(\widetilde{\boldsymbol{x}}_{\boldsymbol{c}}) \cdot f(\widetilde{\boldsymbol{x}}'_{\boldsymbol{c}})}{\|f(\widetilde{\boldsymbol{x}}_{\boldsymbol{c}})\|\|f(\widetilde{\boldsymbol{x}}'_{\boldsymbol{c}})\|} = \frac{\sum_{i=1}^{n} f(\widetilde{\boldsymbol{x}}_{\boldsymbol{c},\boldsymbol{i}}) \, f(\widetilde{\boldsymbol{x}}'_{c,i})}{\sqrt{\sum_{i=1}^{n} f(\widetilde{\boldsymbol{x}}_{\boldsymbol{c},\boldsymbol{i}})^2} \sqrt{\sum_{i=1}^{n} f(\widetilde{\boldsymbol{x}}'_{c,i})^2}}, \tag{4}$$

where $\widetilde{\boldsymbol{x}}_{\boldsymbol{c}}$ and $\widetilde{\boldsymbol{x}}'_{c}$ are two images from the same class $c$, $f(\cdot)$ are the features extracted from a pretrained model, and $n$ is the feature dimension.

### 3.2.2 Similarity between Synthetic and Original Dataset: Maximum Mean Discrepancy

The similarity between images is not the only determinant of diversity since images can be dissimilar to each other yet not representative of the original dataset. Therefore, to further validate the diversity of our synthetic dataset, we consider an additional metric: the Maximum Mean Discrepancy (MMD) between synthetic datasets and original datasets. This measure helps evaluate how well the synthetic data represents the original data distribution. The following proposition clarifies the relationship between MMD and dataset diversity:

**Proposition 2.** *A lower MMD suggests that the synthetic dataset captures a broader range of features similar to the original dataset, indicating greater diversity.*

The empirical approximation of MMD can be formally defined as [28, 29],

$$\text{MMD}^2\left(\boldsymbol{P}_{\mathcal{T}}, \boldsymbol{P}_{\mathcal{S}}\right) = \hat{\mathcal{K}}_{\mathcal{T},\mathcal{T}} + \hat{\mathcal{K}}_{\mathcal{S},\mathcal{S}} - 2\hat{\mathcal{K}}_{\mathcal{T},\mathcal{S}} \tag{5}$$

where $\hat{\mathcal{K}}_{X,Y} = \frac{1}{|X| \cdot |Y|} \sum_{i=1}^{|X|} \sum_{j=1}^{|Y|} \mathcal{K}\left(f\left(x_i\right), f\left(y_j\right)\right)$ with $\{x_i\}_{i-1}^{|X|} \sim X, \{y_i\}_{i=1}^{|Y|} \sim Y$. $\mathcal{T}$ and $\mathcal{S}$ denote real and synthetic datasets, respectively; $\mathcal{K}$ is the reproducing kernel (e.g., Gaussian kernel); $\boldsymbol{P}$ is the feature (embedding) distribution, and $f(\cdot)$ is the feature representation extracted by model $\theta$, where $f(\mathcal{T}) \sim \boldsymbol{P}_{\mathcal{T}}, f(\mathcal{S}) \sim \boldsymbol{P}_{\mathcal{S}}$.

## 3.3 Label Pruning for Large-scale Distillation (LPLD)

### 3.3.1 Diverse Sample Generation via Class-wise Supervision

The previous objective function follows Eq. 3; it uses a subset of classes $\mathcal{B}_c$ to match the BN statistics of the entire dataset, and images in the same class are independently generated, causing an low image diversity within classes. However, inspired by He *et al.*[30], images in the same class should work collaboratively, and images that are optimized individually (see Baseline B in work [30]) do not lead to the optimal performance when IPC (Images Per Class) gets larger.

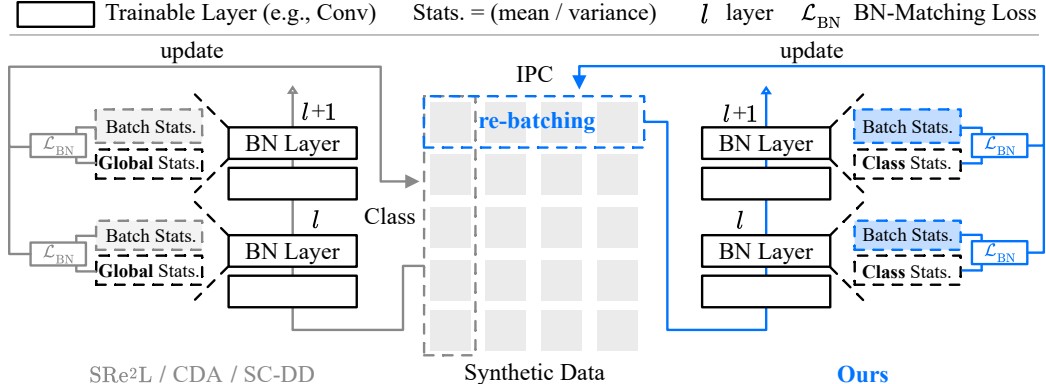

Figure 4: Illustration of existing methods (left, grey) and the proposed method (right, blue). Existing methods (i.e., SRe$^2$L, CDA) independently generate along the IPC (Image-Per-Class) dimension, causing a high similarity between images of the same class. The proposed method allows images of the same class to collaborate, leaving different classes naturally independent. In addition, synthetic images are updated under class-wise supervision. The classification loss is omitted for simplicity.

**Step 1: Re-batching Images within Class.** Subsequently, to obtain a collaborative effect among different images of the same class, we sample images from the same class and provide the images with class-wise supervision [4, 24]. Fig. 4 illustrates the changes.

**Step 2: Introducing Class-wise Supervision.** However, the running mean and variance approximate the original dataset's expected mean and variance in a global aspect. The matching objective becomes sub-optimal in class-wise matching situation. To this end, we propose to track BN statistics for each class separately. Since we only track the running mean and variance, the extra storage is marginal even when up to 1K classes in ImageNet-1K (see Appendix E.2 and E.4).

**Step 3: Class-wise Objective Function.** The new class-wise objective function is modified from Eq. 3, which has two loss functions. First, we compute the classification loss (i.e., the Cross-Entropy Loss) with BN layers using global statistics to ensure effective supervision. Second, we compute BN loss by matching class-wise BN statistics. The modified parts are highlighted in blue color, and the objective function is formally put as,

$$
\underset{\widetilde{\boldsymbol{x}}_c}{\arg\min}\left(\overbrace{-\sum_{i=1}^{N} y_{c,i} \log\left(\text{softmax}\left(\boldsymbol{\theta}_{\mathcal{T}}\left(\overbrace{\frac{\widetilde{\boldsymbol{x}}_{c,i} - \mathbf{BN}_{\text{global}}^{\text{RM}}}{\sqrt{\mathbf{BN}_{\text{global}}^{\text{RV}} + \epsilon}}}\right)\right)\right)_c}^{\text{Cross-Entropy Loss with Global BN Statistics}}\right.
$$
$$
\left. + \alpha \cdot \underbrace{\sum_{l}\left(\left\|\mu_l(\widetilde{\boldsymbol{x}}_c) - \mathbf{BN}_{l,c}^{\text{RM}}\right\|_2 + \left\|\sigma_l^2(\widetilde{\boldsymbol{x}}_c) - \mathbf{BN}_{l,c}^{\text{RV}}\right\|_2\right)}_{\text{Batch Norm Loss with Class-wise BN Statistics}}\right)
\tag{6}
$$

We want to emphasize that even though we are adjusting the BN loss with class-wise statistics, the global statistics of the dataset are still taken into account. The output logits for calculating CE loss are produced using global statistics. This is because altering $\mu$ and $\sigma$ without fine-tuning $\gamma$ and $\beta$ could lead to a decline in model performance, resulting in less effective supervision.

**Theoretical Number of Updates for Stable Class-wise BN Statistics.** Traditional BN layers do not compute class-wise statistics; therefore, we need to either keep track of the class-wise statistics while training a model from scratch or compute these statistics using a pretrained model. We prefer the latter as the former requires extensive computing resources. To understand how many BN statistics updates are needed, we can first look at the update rules of BN running statistics for a class $c$:

$$
\mathbf{BN}_{l,c}^{\text{RM}} \leftarrow (1-\epsilon) \cdot \mathbf{BN}_{l,c}^{\text{RM}} + \epsilon \cdot \mu_l(\boldsymbol{x}_c),
$$
$$
\mathbf{BN}_{l,c}^{\text{RV}} \leftarrow (1-\epsilon) \cdot \mathbf{BN}_{l,c}^{\text{RV}} + \epsilon \cdot \sigma_l^2(\boldsymbol{x}_c),
\tag{7}
$$

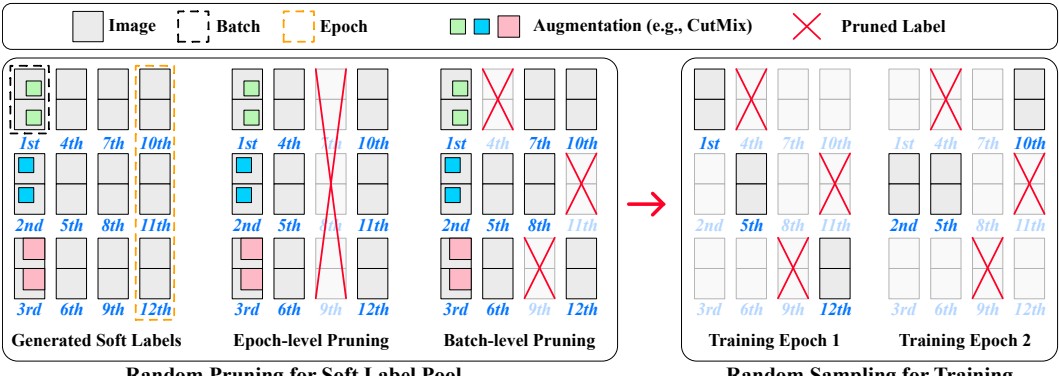

Figure 5: Illustration of two random processes in label pruning with improved label pool. First, we need a smaller soft label pool due to the storage budget. We can conduct pruning at two levels: (1) epoch-level and (2) batch-level. Batch-level pruning can provide a more diverse label pool since augmentations (e.g., Mixup or CutMix) are different across batches. The illustrated pruning ratio is 25%; the crossed-out labels denote the pruned labels, and the remaining form the label pool. Second, we randomly sample soft labels for model training.

where $\epsilon$ is the momentum. Since the momentum factor for the current batch statistics is usually set to a small value (i.e., $\epsilon = 0.1$), we can theoretically compute existing running statistics that can be statistically significant after how many updates, assuming all other factors are fixed.

Since the running statistics are computed per class, we provide the theoretical number of updates required to stabilize all class statistics (see Appendix A for the proof):

$$n \geq \max \left( \underbrace{\frac{-2\ln\left(\frac{T}{2}\right)}{\delta^2 \min(q_c)}}_{\text{Chernoff Bound}}, \quad \underbrace{\frac{\ln\left(\frac{C}{\tau}\right)}{(1-\delta)\varepsilon \min(q_c)}}_{\text{BN Convergence}} \right). \tag{8}$$

where $n$ is the number updates needed, $q_c$ is the probability that class $c$ appears in a batch, $T$ is a probability threshold, $\varepsilon$ is the momentum parameter in Batch Normalization, $\delta$ is the acceptable relative deviation (where $0 \leq \delta \leq 1$), $C$ is some constant, and $\tau$ is the desired convergence tolerance for the BN statistics. How Eq. 8 guides our experiment design is detailed in Appendix E.3.

### 3.3.2 Random Label Pruning with Improved Label Pool

**Excelling in Both Similarity Measures.** By adopting the changes provided in Sec. 3.3.1, our synthetic dataset is more diverse and representative than the existing methods. First, our dataset exhibits smaller feature cosine similarity within classes compared to datasets produced by existing methods, as shown in Table 1. This indicates that our synthetic images are less similar to each other and, thus, more diverse. Second, our dataset exhibits a significantly lower MMD shown in Fig 3 compared to datasets produced by existing methods. This suggests that our synthetic dataset better captures the feature distribution of the original dataset. After obtaining a diverse dataset, the next move is to address superfluous soft labels.

**Random Label Pruning.** Different from dataset pruning metrics, which many wield training dynamics [31, 32], label pruning is inherently different since the labels in different epochs are independently generated or evaluated. Subsequently, these methods do not directly apply, and we modify these metrics to determine which epochs contain the most useful augmentations and soft labels. Through empirical study, we find that using soft labels carefully pruned from different metrics is **no better** than simple random pruning. As a result, we can discard complex rule-based pruning metrics, attaining both simplicity and efficiency. After obtaining the soft label pool, we have to decide which labels will be used. Following the previous random pruning scheme, we randomly sample the labels for model training in order to ensure diversity and avoid any prior knowledge.

**Improved Label Pool.** Considering that random selection may be the most efficient choice, we rethink the diversity of the label pool, as labels at the epoch-level are not the finest elements.

Table 2: Tiny-ImageNet label pruning results. The standard deviation is attained from three different runs. $^\dagger$ denotes the reported results.

(a) Comparison between SOTA methods.

| ResNet-18 | 1× | | | 10× | | | 20× | | | 30× | | | 40× | | |
|---|---|---|---|---|---|---|---|---|---|---|---|---|---|---|---|
| | SRe²L | CDA | Ours | SRe²L | CDA | Ours | SRe²L | CDA | Ours | SRe²L | CDA | Ours | SRe²L | CDA | Ours |
| IPC50 | 41.1$^\dagger$ | 48.7$^\dagger$ | **48.8**±0.4 | 40.3 | 45.0 | **46.7**±0.6 | 39.0 | 41.2 | **44.3**±0.5 | 34.6 | 35.8 | **40.2**±0.3 | 29.8 | 30.9 | **38.4**±1.3 |
| IPC100 | 49.7$^\dagger$ | 53.2$^\dagger$ | **53.6**±0.3 | 48.3 | 50.7 | **52.2**±0.2 | 46.5 | 48.0 | **50.6**±0.2 | 43.0 | 44.2 | **47.6**±0.2 | 39.4 | 40.0 | **46.1**±0.2 |

(b) Experiments on larger networks.

| | 1× | | 10× | | 20× | | 30× | | 40× | |
|---|---|---|---|---|---|---|---|---|---|---|
| | ResNet-50 | ResNet-101 | ResNet-50 | ResNet-101 | ResNet-50 | ResNet-101 | ResNet-50 | ResNet-101 | ResNet-50 | ResNet-101 |
| IPC50 | **49.0**±1.6 | **49.7**±0.9 | **48.6**±0.5 | **48.7**±0.5 | **47.2**±0.6 | **46.4**±0.6 | **43.4**±0.0 | **43.0**±0.9 | **42.3**±0.2 | **42.1**±1.2 |
| IPC100 | **55.3**±0.2 | **55.4**±0.4 | **54.0**±0.3 | **54.1**±0.8 | **52.7**±0.3 | **53.7**±0.4 | **51.0**±0.5 | **51.1**±0.2 | **50.3**±0.5 | **48.7**±1.7 |

Table 3: ImageNet-1K label pruning result. Our method consistently shows a better performance under various pruning ratios. The validation model is ResNet-18. $^\dagger$ denotes the reported results.

| ResNet-18 | 1× | | | 10× | | | 20× | | | 30× | | | 40× | | |
|---|---|---|---|---|---|---|---|---|---|---|---|---|---|---|---|
| | SRe²L | CDA | Ours | SRe²L | CDA | Ours | SRe²L | CDA | Ours | SRe²L | CDA | Ours | SRe²L | CDA | Ours |
| IPC10 | 20.1 | 33.3 | **34.6**±0.9 | 18.9 | 28.4 | **32.7**±0.6 | 16.0 | 21.9 | **28.6**±0.4 | 14.1 | 14.2 | **23.1**±0.1 | 11.4 | 13.2 | **20.2**±0.3 |
| IPC20 | 33.6 | 44.0 | **47.2**±0.5 | 31.1 | 39.7 | **44.7**±0.4 | 29.2 | 34.1 | **41.0**±0.3 | 24.5 | 27.5 | **35.9**±0.3 | 21.7 | 24.0 | **33.0**±0.6 |
| IPC50 | 46.8$^\dagger$ | 53.5$^\dagger$ | **55.4**±0.3 | 44.1 | 50.3 | **54.4**±0.2 | 41.5 | 46.1 | **51.8**±0.2 | 37.2 | 41.8 | **48.6**±0.2 | 35.5 | 38.0 | **46.7**±0.3 |
| IPC100 | 52.8$^\dagger$ | 58.0$^\dagger$ | **59.4**±0.2 | 51.1 | 55.1 | **58.8**±0.0 | 49.5 | 53.3 | **57.4**±0.0 | 46.7 | 49.7 | **55.2**±0.1 | 44.4 | 47.2 | **54.0**±0.8 |
| IPC200 | 57.0$^\dagger$ | 63.3$^\dagger$ | **62.6**±0.3 | 56.5 | 59.4 | **62.4**±0.7 | 55.1 | 58.3 | **61.7**±0.7 | 52.9 | 56.0 | **60.1**±0.5 | 51.9 | 54.4 | **59.6**±0.6 |

The augmentations such as CutMix and Mixup are performed at the batch level, where the same augmentations are applied to images within the same batch and are different across batches. Therefore, we improve the label pool by allowing batches in different epochs to form a new epoch. The improved label pool breaks the fixed batch orders and the fixed combination of augmentations within an epoch, allowing a more diverse training process while reusing the labels. Our label pruning method is illustrated in Fig. 5.

# 4 Experiments

## 4.1 Experiment Settings

Dataset details can be found in Appendix B and detailed settings are provided in Appendix C. Computing resources used for experiments can be found in Appendix E.5.

**Dataset.** Our experiment results are evaluated on Tiny-ImageNet [33], ImageNet-1K [34], and ImageNet-21K-P [35]. We follow the data pre-processing procedure of SRe²L [6] and CDA [7].

**Squeeze.** We modify the pretrained model by adding class-wise BN running mean and running variance; since they are not involved in computing the BN statistics, they do not affect performance. As mentioned in Sec. 3.3.1, we compute class-wise BN statistics by training for one epoch with model parameters kept frozen.

**Recover.** We perform data synthesis following Eq. 6. The batch size for the recovery phase is the same as the IPC. Besides, we adhere to the original setting in SRe²L.

**Relabel.** We use pretrained ResNet18 [36] for all experiments as the relabel model except otherwise stated. For Tiny-ImageNet and ImageNet-1K, we use Pytorch pretrained model. For ImageNet-21K-P, we use Timm pretrained model.

**Validate.** For validation, we adhere to the hyperparameter settings of CDA [7].

**Pruning Setting.** For label pruning, we exclude the last batch (usually with an incomplete batch size) of each epoch from the label pool. There are two random processes: (1) Random candidate selection from all batches. (2) Random reuse of candidate labels.

Table 5: Comparison between different pruning metrics. Results are obtained from ImageNet-1K IPC10 and validated using ResNet-18.

(b) Calibration of label pool.

(a) Random pruning vs. Pruning metrics at 40×.

| IPC10 | correct | diff | diff_signed | cut_ratio | confidence |
|---|---|---|---|---|---|
| Hard | 19.6 | 18.9 | 19.2 | 19.5 | 19.0 |
| Easy | 19.3 | 18.7 | 19.3 | 19.5 | 17.9 |
| Uniform | 20.0 | 18.5 | 20.1 | 19.7 | 19.2 |
| Random | | | 20.2 | | |

| Easy | Hard | 20× | 30× | 50× | 100× |
|---|---|---|---|---|---|
| 0 | -90% | 25.5 | 22.8 | 17.4 | 10.3 |
| 0 | -50% | 28.2 | 22.5 | 17.8 | 9.7 |
| -10% | -30% | 28.3 | 22.6 | 17.1 | 8.7 |
| -30% | -5% | 27.8 | 21.5 | 16.0 | 7.9 |
| -90% | 0 | 27.7 | 22.1 | 16.3 | 8.6 |
| 0 | 0 | 28.6 | 23.1 | 17.6 | 9.6 |

## 4.2 Primary Result

**Tiny-ImageNet.** Table 2a presents a comparison between the label pruning outcomes on Tiny-ImageNet for our approach, SRe$^2$L [6], and the subsequent work, CDA [7]. Our method not only consistently surpasses SRe$^2$L across identical pruning ratios but also achieves comparable results to SRe$^2$L while using 40× fewer labels. When compared to CDA, our method exhibits closely matched performance, yet it demonstrates superior accuracy preservation. For instance, at a 40× label reduction, our method secures a notable 7.5% increase in accuracy over CDA, even though the improvement stands at a mere 0.1% at the 1× benchmark. Table 2b provides the pruning results on ResNet50 and ResNet101. Although there are consistent improvements observed when compared to ResNet18, scaling to large networks does not necessarily bring improvements.

**ImageNet-1K.** Table 3 compares the ImageNet-1K pruning results with SOTA methods on ResNet18. Our method outperforms other SOTA methods at various pruning ratios and different IPCs. More importantly, our method consistently exceeds the unpruned version of SRe$^2$L with 30× less storage. Such a result is not impressive at first glance; however, when considering the actual storage, the storage is reduced from 29G to 0.87G. In addition, we notice the performance at 10× (or 90%) pruning ratio degrades slightly, especially for large IPCs. For example, merely 0.2% performance degradation on IPC200 using ResNet18. Pruning results of larger IPCs can be found in Appendix D.2.

## 4.3 Analysis

**Ablation Study.** Table 4 presents the ablation study of the proposed method. **Row 1** is the implementation of SRe$^2$L under CDA's hyperparameter settings. **Row 2** is simply re-ordering the loops, and the performance at 1× is improved; nevertheless, when considering the extreme pruning ratio (i.e., 100×), it falls short of the existing method. **Row 3** computes class-wise BN running statistics in the "squeeze" phase,

Table 4: Ablation study of the proposed method. `C` denotes using class-wise matching. `CS` denotes suing class-wise supervision. `ILP` denotes using an improved label pool. (IPC50, ResNet18, ImageNet-1K).

| +C | +CS | +ILP | 1× | 10× | 20× | 30× | 50× | 100× |
|---|---|---|---|---|---|---|---|---|---|
| - | - | - | 52.0 | 49.4 | 46.4 | 41.1 | 34.8 | 25.4 |
| ✓ | - | - | 54.7 | 51.9 | 48.5 | 42.9 | 37.7 | 22.6 |
| ✓ | ✓ | - | 55.3 | 53.2 | 49.9 | 45.7 | 39.7 | 29.1 |
| ✓ | ✓ | ✓ | 55.4 | 54.4 | 51.8 | 48.6 | 43.1 | 33.7 |

and these class-wise statistics are used as supervision in the "recover" phase. A steady improvement is observed. **Row 4** allows pre-generated labels to be sampled at batch level from different epochs, further boosting the performance. Refer to Appendix D.1 for an expanded version of ablation.

**Label Pruning Metrics.** From Table 5a, we empirically find that using different metrics explained in Appendix E.1 is **no better than** random pruning. In addition, as mentioned in FerKD [25], calibrating the searching space by discarding a portion of easy or hard images can be beneficial. We conduct a similar experiment to perform random pruning on a calibrated label pool, and the metric for determining easy or hard images is "confidence". However, as shown in Table 5b, no such range can consistently outperform the non-calibrated ones (last row). An interesting observation is that the label pruning law **at large pruning ratio** seems to coincide partially with data pruning, where removing hard labels becomes beneficial [37].

**Generalization.** Table 6a shows the performance under large compression rates. Smaller IPC datasets suffer more from label pruning since it requires more augmentation and soft label pairs to boost data diversity. Furthermore, label pruning results on ResNet50 are provided in Table 6b.

Table 6: Additional ImageNet-1K label pruning results.

(a) Large pruning rate.

| ResNet-18 | 50× | 100× |
|---|---|---|
| IPC10 | 17.6 | 9.6 |
| IPC20 | 30.0 | 17.9 |
| IPC50 | 43.1 | 33.7 |
| IPC100 | 52.0 | 44.7 |
| IPC200 | 57.7 | 52.6 |

(b) Label pruning results on ResNet-50.

| ResNet-50 | 1× | 10× | 20× | 30× | 50× | 100× |
|---|---|---|---|---|---|---|
| IPC10 | 41.7 | 37.7 | 35.4 | 27.5 | 22.6 | 11.0 |
| IPC20 | 54.4 | 52.3 | 48.9 | 45.4 | 39.5 | 24.0 |
| IPC50 | 62.2 | 61.2 | 58.8 | 56.2 | 52.3 | 44.7 |
| IPC100 | 65.7 | 65.1 | 63.9 | 62.0 | 59.8 | 54.2 |
| IPC200 | 67.8 | 67.1 | 66.7 | 65.4 | 64.1 | 60.1 |

(c) Cross-architecture result. IPC50.

| Model | Size | Full Acc | 1× | 10× | 30× |
|---|---|---|---|---|---|
| ResNet-18 [36] | 11.7M | 69.76 | 55.44 | 54.45 | 48.62 |
| ResNet-50 [36] | 25.6M | 76.13 | 62.24 | 61.22 | 56.24 |
| EfficientNet-B0 [38] | 5.3M | 77.69 | 55.51 | 54.69 | 52.10 |
| MobileNet-V2 [39] | 3.5M | 71.88 | 49.12 | 49.26 | 45.80 |
| Swin-V2-Tiny [40] | 28.4M | 82.07 | 40.59 | 37.35 | 29.54 |

Table 7: Label pruning result on ImageNet-21K-P, using ResNet-18. $\mathbb{I}$ denotes image storage. $\mathbb{L}$ denotes label storage. † denotes reported results.

| IPC | $\mathbb{I}$ | $\mathbb{L}$ | 1× SRe²L | CDA | Ours | $\mathbb{L}$ | 10× Ours | $\mathbb{L}$ | 40× Ours |
|---|---|---|---|---|---|---|---|---|---|
| IPC10 | 3G | 643G | 18.5† | 22.6† | 25.4 | 65G | 24.1 | 16G | 21.3 |
| IPC20 | 5G | 1285G | 20.5† | 26.4† | 30.3 | 129G | 31.3 | 32G | 29.4 |

Table 8: Label pruning for optimization-free method. "Ours" uses improved label pool.

| ResNet-18 | 10× RDED | Ours | 20× RDED | Ours | 30× RDED | Ours | 40× RDED | Ours |
|---|---|---|---|---|---|---|---|---|
| IPC10 | 37.9 | 39.1 | 32.5 | 35.7 | 25.4 | 30.8 | 24.0 | 29.1 |
| IPC20 | 45.8 | 48.1 | 41.2 | 44.3 | 36.2 | 39.5 | 32.9 | 38.4 |
| IPC50 | 53.2 | 54.3 | 49.9 | 52.7 | 48.8 | 49.7 | 44.3 | 48.7 |
| IPC100 | 57.3 | 57.8 | 55.3 | 57.1 | 55.2 | 55.3 | 51.4 | 54.2 |

Not only scaling to large networks of the same family (i.e., ResNet) but Table 6c also demonstrates the generalization capability of the proposed method across different network architectures. An analogous trend is evident in the context of label pruning: comparable performance is achieved with 10× fewer labels. This reinforces the statement that the necessity for extensive augmentations and labels can be significantly reduced if the dataset exhibits sufficient diversity.

**Large Dataset.** ImageNet-21K-P has 10,450 classes, significantly increasing the disk storage as each soft label stores a probability of 10,450 classes. The IPC20 dataset leads to a 1.2 TB (i.e., 1285 GB) label storage, making the existing framework less practical. However, with the help of our method, it can surpass SRe²L [6] by a large margin despite using 40× less storage. For example, we attain an 8.9% accuracy improvement on IPC20 with label storage reduced from 1285 GB to 32 GB.

**Pruning for Optimization-Free Approach.** RDED [8] is an optimization-free approach during the "recover" phase. However, extensive labels are still required for post-evaluation. To prune labels, consistent improvements are observed using the improved label pool, as shown in Table 8.

**Comparison with G-VBSM [9].** Compared to G-VBSM [9], which uses an ensemble of 4 models to recover and relabel, our method outperforms it at various pruning ratios with only a single model (see Table 9). Furthermore, the techniques used for G-VBSM apply to our method. By adopting label generation with ensemble and a loss function of "MSE+0.1 × GT" [9], our method can be further improved by a large margin on IPC10 of ImageNet-1K, using ResNet18. Implementation details can be found in Appendix C.4.

Table 9: Compare with G-VBSM [9]. "Ours+" uses ensemble and MSE+GT loss.

| IPC10 | G-VBSM | Ours | Ours+ |
|---|---|---|---|
| 1× | 31.4 | 35.7 | 39.0 |
| 10× | 28.4 | 32.7 | 37.6 |
| 20× | 26.5 | 28.6 | 34.8 |
| 30× | 22.5 | 23.1 | 30.3 |
| 40× | 18.8 | 20.2 | 27.9 |

**Visualization.** Fig. 2b visualizes our method on three classes. More visualizations are provided in Appendix F.

## 5 Conclusion

To answer the question *"whether large-scale soft labels are necessary for large-scale dataset distillation?"*, we conduct diversity analysis on synthetic datasets. The high within-class similarity is observed and necessitates large-scale soft labels. Our LPLD method re-batches images within classes and introduces class-wise BN supervision during the image synthesis phase to address this issue. These changes improve data diversity, so that simple random label pruning can perform on par with complex rule-based pruning metrics. Additionally, we randomly conduct pruning on an improved label pool. Finally, LPLD is validated by extensive experiments, serving a strong baseline that takes into account actual storage. Limitations and future works are provided in Appendix E.6. The ethics statement and broader impacts can be found in Appendix E.7.

## Acknowledgement

This work was supported in part by A*STAR Career Development Fund (CDF) under C233312004, in part by the National Research Foundation, Singapore, and the Maritime and Port Authority of Singapore / Singapore Maritime Institute under the Maritime Transformation Programme (Maritime AI Research Programme – Grant number SMI-2022-MTP-06). The computational work for this article was partially performed on resources of the National Supercomputing Centre (NSCC), Singapore (`https://www.nscc.sg`).

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

# A  Proof

We aim to determine a lower bound on the number of batches (updates) $n$ required to ensure that the Batch Normalization (BN) statistics for each class in the ImageNet dataset converge within a specified tolerance $\tau$, with high probability. The dataset has a varying number of images per class, affecting the probability of each class appearing in a batch during sampling.

## A.1  Preliminary Analysis

**Defining Class Probabilities:** Let $p_c$ denote the probability that a randomly selected image from the dataset belongs to class $c$:

$$p_c = \frac{\text{Number of images in class } c}{\text{Total number of images in the dataset}}.$$

Due to the unequal distribution of images across classes, $p_c$ varies among classes.

**Probability of Class Appearance in a Batch:** When sampling a batch of size $B$, the probability that class $c$ does not appear in the batch is $(1 - p_c)^B$. Therefore, the probability that class $c$ appears in the batch is:

$$q_c = 1 - (1 - p_c)^B.$$

This represents the likelihood that at least one image from class $c$ is included in a given batch.

**Number of Batches:** Let $n$ be the total number of batches sampled during training.

We assume that batches are sampled independently with replacement from the dataset. Under this assumption, each batch is an independent trial where class $c$ appears with probability $q_c$. Therefore, the number of batches $M$ where class $c$ appears follows a binomial distribution:

$$M \sim \text{Binomial}(n, q_c).$$

**Remark:** In practice, batches are often sampled without replacement within an epoch, introducing dependency between batches. However, for large datasets where the total number of images $N$ is significantly larger than the batch size $B$ and the number of batches $n$, the dependence becomes negligible. In such cases, the binomial distribution serves as a reasonable approximation.

The expected value of $M$ is:

$$E[M] = nq_c.$$

## A.2  Chernoff Bound

To ensure that $M$ is not significantly less than its expected value $E[M]$, we apply the Chernoff bound:

$$\Pr\left(M \leq (1 - \delta)E[M]\right) \leq \exp\left(-\frac{\delta^2 E[M]}{2}\right),$$

where $\delta \in (0, 1)$ represents the acceptable relative deviation from the expected value. This bound provides a way to quantify the probability that a random variable deviates from its expected value, which is crucial for making high-confidence guarantees.

To ensure the probability that $M$ is less than $(1 - \delta)E[M]$ is at most $T_1$, we set:

$$\exp\left(-\frac{\delta^2 nq_c}{2}\right) \leq T_1.$$

Solving for $n$:

$$n \geq \frac{-2\ln(T_1)}{\delta^2 q_c}.$$

To ensure this condition holds for all classes, we use the minimum value of $q_c$:

$$n \geq \frac{-2\ln(T_1)}{\delta^2 \min(q_c)}.$$

### A.3 BN Convergence

**BN Statistics Update:** Batch Normalization updates its running statistics using an exponential moving average. The update rule for the BN statistics of class $c$ at iteration $t+1$ is:

$$\text{BN}_c^{t+1} = (1-\varepsilon)\text{BN}_c^t + \varepsilon\hat{\text{BN}}_c^{t+1},$$

where $\varepsilon$ is the momentum parameter, and $\hat{\text{BN}}_c^{t+1}$ is the BN statistics estimated from the current batch for class $c$.

Since BN statistics for class $c$ are updated only when class $c$ appears in the batch, we consider only the updates corresponding to those batches. After $M$ such updates:

$$\text{BN}_c^{t+M} = (1-\varepsilon)^M \text{BN}_c^t + \varepsilon\sum_{k=1}^{M}(1-\varepsilon)^{M-k}\hat{\text{BN}}_c^k,$$

where $\hat{\text{BN}}_c^k$ is the BN statistics estimated in the $k$-th batch containing class $c$.

Assuming $\hat{\text{BN}}_c^k$ is an unbiased estimator of the true BN statistics $\mu_c$ when class $c$ appears, the expected value is:

$$E[\text{BN}_c^{t+M}] = (1-\varepsilon)^M \text{BN}_c^t + \mu_c\left[1 - (1-\varepsilon)^M\right].$$

**Convergence Within Tolerance:** To ensure that the BN statistics converge within a tolerance $\tau$ of the true statistics $\mu_c$:

$$\left|E[\text{BN}_c^{t+M}] - \mu_c\right| \leq \tau.$$

Since:

$$\left|E[\text{BN}_c^{t+M}] - \mu_c\right| = (1-\varepsilon)^M \left|\text{BN}_c^t - \mu_c\right|,$$

and assuming $\left|\text{BN}_c^t - \mu_c\right| \leq C$ for some constant $C$, we have:

$$(1-\varepsilon)^M C \leq \tau.$$

Taking natural logarithms:

$$M\ln(1-\varepsilon) + \ln(C) \leq \ln(\tau).$$

Using the approximation $\ln(1-\varepsilon) \approx -\varepsilon$ for small $\varepsilon$:

$$-M\varepsilon + \ln(C) \leq \ln(\tau).$$

Solving for $M$:

$$M \geq M_0 = \frac{\ln\left(\frac{C}{\tau}\right)}{\varepsilon}.$$

**Origin of $M_0$:** Here, $M_0$ is derived from the BN convergence requirement that ensures:

$$(1-\varepsilon)^{M_0} \left|\text{BN}_c^t - \mu_c\right| \leq \tau.$$

It represents the minimum number of updates required for the BN statistics of class $c$ to converge within the desired tolerance $\tau$.

### A.4 Combining Bounds

**Event Definitions:**

- Let $E_1$ be the event that class $c$ appears in sufficient batches (as guaranteed by the Chernoff bound).
- Let $E_2$ be the event that the BN statistics for class $c$ converge within the desired tolerance $\tau$.

**Target Probability:** We aim to ensure that both events occur simultaneously with high probability:
$$P(E_1 \cap E_2) \geq 1 - T.$$

**Union Bound Application:** For any two events, the probability of their intersection satisfies:
$$
\begin{aligned}
P(E_1 \cap E_2) &= 1 - P(\overline{E_1 \cap E_2}) \\
&= 1 - P(\overline{E_1} \cup \overline{E_2}) \\
&\geq 1 - P(\overline{E_1}) - P(\overline{E_2}).
\end{aligned}
$$

**Error Probability Allocation:** For simplicity, we allocate the total acceptable failure probability $T$ equally between the two events:

$$P(\overline{E_1}) \leq \frac{T}{2} \quad \text{(allocated to Chernoff bound)},$$

$$P(\overline{E_2}) \leq \frac{T}{2} \quad \text{(allocated to BN convergence)}.$$

**Chernoff Bound Analysis:** For event $E_1$, we require that the probability of class $c$ appearing in fewer than the expected number of batches is at most $\frac{T}{2}$:

$$P\left(M \leq (1-\delta)nq_c\right) \leq \frac{T}{2}.$$

Applying the Chernoff bound:
$$\exp\left(-\frac{\delta^2 nq_c}{2}\right) \leq \frac{T}{2}.$$

Solving for $n$:
$$-\frac{\delta^2 nq_c}{2} \leq \ln\left(\frac{T}{2}\right),$$
$$n \geq \frac{-2\ln\left(\frac{T}{2}\right)}{\delta^2 q_c}.$$

**BN Convergence Requirement:** For event $E_2$, we require that the number of batches $M$ where class $c$ appears is sufficient for BN convergence:

$$M \geq M_0 = \frac{\ln\left(\frac{C}{\tau}\right)}{\varepsilon}.$$

To ensure that this condition holds when event $E_1$ occurs, we use the fact that, with probability at least $1 - \frac{T}{2}$, we have:

$$M \geq (1 - \delta)nq_c.$$

Therefore, to guarantee $M \geq M_0$, we require:

$$(1 - \delta)nq_c \geq M_0 = \frac{\ln\left(\dfrac{C}{\tau}\right)}{\varepsilon}.$$

Solving for $n$:

$$n \geq \frac{\ln\left(\dfrac{C}{\tau}\right)}{(1 - \delta)\varepsilon q_c}.$$

**Final Combined Bound:** To ensure that both conditions hold for all classes, we use $\min(q_c)$:

$$n \geq \max\left(\underbrace{\frac{-2\ln\left(\dfrac{T}{2}\right)}{\delta^2 \min(q_c)}}_{\text{Chernoff Bound}}, \quad \underbrace{\frac{\ln\left(\dfrac{C}{\tau}\right)}{(1 - \delta)\varepsilon \min(q_c)}}_{\text{BN Convergence}}\right),$$

where $\delta$ represents the acceptable relative deviation from the expected number of batches, $\varepsilon$ is the momentum parameter in BN updates, $T$ denotes the acceptable total failure probability ($T = T_1 + T_2$), $\tau$ is the convergence threshold for BN statistics, $C$ represents an upper bound on $\left|\text{BN}_c^t - \mu_c\right|$ at initialization, and $\min(q_c)$ represents the minimum probability that a class appears in a batch.

This bound ensures:

- With probability at least $1 - \frac{T}{2}$, each class $c$ appears in at least $(1 - \delta)nq_c$ batches (event $E_1$ occurs).
- With probability at least $1 - \frac{T}{2}$, the BN statistics for each class $c$ converge within tolerance $\tau$ (event $E_2$ occurs).
- By the union bound, both events $E_1$ and $E_2$ occur simultaneously with probability at least $1 - T$.

## B  Dataset Details

We perform experiments on the following three datasets:

- Tiny-ImageNet [33] is the subset of ImageNet-1K containing 500 images per class of a total of 200 classes, and spatial sizes of images are downsampled to $64 \times 64$.
- ImageNet-1K [34] contains 1,000 classes and 1,281,167 images in total. The image sizes are resized to $224 \times 224$.
- ImageNet-21K-P [35] is the pruned version of ImageNet-21K, containing 10,450 classes and 11,060,223 images in total. Images are sized to $224 \times 224$ resolution.

## C  Hyperparameter Settings

### C.1  ImageNet-1K

Table 10: Squeezing and class-wise BN statistics of ImageNet-1K.

| Info | Detail |
|---|---|
| Total Images | 1,281,167 |
| Batch Size | 256 |
| BN Updates | 5005 |
| Source | https://github.com/pytorch/vision/tree/main/references/classification |

Table 11: Data Synthesis of ImageNet-1K.

| Config | Value | Detail |
|---|---|---|
| Iteration | 4,000 | - |
| Optimizer | Adam | $\beta_1, \beta_2 = (0.5, 0.9)$ |
| Image LR | 0.25 | - |
| Batch Size | IPC-dependent | e.g., 50 for IPC50 |
| Initialization | Random | - |
| BN Loss ($\alpha$) | 0.01 | - |

Table 12: Relabel and Validation of ImageNet-1K.

| Config | Value | Detail |
|---|---|---|
| Epochs | 300 | - |
| Optimizer | AdamW | - |
| Model LR | 0.001 | - |
| Batch Size | 128 | - |
| Scheduler | CosineAnnealing | - |
| EMA Rate | Not Used | - |
| | RandomResizedCrop | scale ratio = (0.08, 1.0) |
| Augmentation | RandomHorizontalFlip | probability = 0.5 |
| | CutMix | - |

We use Pytorch pretrained ResNet-18 [36], with a Top-1 accuracy of 69.76%, as both the recovery and relabeling model. Class-wise BN statistics are computed using a modified version of the training script of the source provided in Table 10. The recovery, or data synthesis, phase is provided in Table 11, which follows CDA [7] except by changing the batch size to an IPC-dependent size. Relabel and validation processes share the same setting as provided in Table 12.

## C.2 Tiny-ImageNet

Table 13: Squeezing and class-wise BN statistics of Tiny-Imagenet.

| Info | Detail |
|---|---|
| Total Images | 100,000 |
| Batch Size | 256 |
| BN Updates | 391 |
| Source | https://github.com/zeyuanyin/tiny-imagenet |

Table 14: Data Synthesis of Tiny-ImageNet.

| Config | Value | Detail |
|---|---|---|
| Iteration | 4,000 | - |
| Optimizer | Adam | $\beta_1, \beta_2 = (0.5, 0.9)$ |
| Image LR | 0.1 | - |
| Batch Size | IPC-dependent | e.g., 50 for IPC50 |
| Initialization | Random | - |
| BN Loss ($\alpha$) | 0.05 | - |

Table 15: Relabel and Validation of Tiny-ImageNet.

| Config | Value | Detail |
|---|---|---|
| Epochs | 100 | - |
| Optimizer | SGD | $\rho = 0.9, \epsilon = 0.0001$ |
| Model LR | 0.2 | - |
| Batch Size | 64 | - |
| Warm-up Scheduler | Linear | epoch = 5, $\epsilon = 0.01$ |
| Scheduler | CosineAnnealing | - |
| EMA Rate | Not Used | - |
| Augmentation | RandomResizedCrop | scale ratio = (0.08, 1.0) |
| | RandomHorizontalFlip | probability = 0.5 |

Following SRe$^2$L and CDA [7], we use a modified version of ResNet-18 [41] for Tiny-ImageNet. We modify the training script from Table 13 to compute class-wise BN statistics. The pretrained model has a Top-1 accuracy of 59.47%, and the model is used for data synthesis and relabel/validation as shown in Table 14 and Table 15, respectively. Note that for the validation phase, a warm-up of 5 epochs is added with a different learning rate scheduler (i.e., linear).

## C.3 ImageNet-21K-P

Table 16: Squeezing and class-wise BN statistics of Imagenet-21K-P.

| Info | Detail |
|---|---|
| Total Images | 11,060,223 |
| Batch Size | 1,024 |
| BN Updates | 10,801 |
| Source | https://github.com/Alibaba-MIIL/ImageNet21K |

Table 17: Data Synthesis of ImageNet-21K-P.

| Config | Value | Detail |
|---|---|---|
| Iteration | 2,000 | - |
| Optimizer | Adam | $\beta_1, \beta_2 = (0.5, 0.9)$ |
| Image LR | 0.05 | - |
| Batch Size | IPC-dependent | e.g., 20 for IPC20 |
| Initialization | Random | - |
| BN Loss ($\alpha$) | 0.25 | - |

Table 18: Relabel and Validation of ImageNet-21K-P.

| Config | Value | Detail |
|---|---|---|
| Epochs | 300 | - |
| Optimizer | AdamW | decay = 0.01 |
| Model LR | 0.002 | - |
| Batch Size | 32 | - |
| Scheduler | CosineAnnealing | - |
| Label Smoothing | 0.2 | - |
| EMA Rate | Not Used | - |
| Augmentation | RandomResizedCrop | scale ratio = (0.08, 1.0) |
| | CutOut | - |

Following CDA [7], we use ResNet-18 trained for 80 epochs initialized with well-trained ImageNet-1K weight [35]. Class-wise BN statistics are computed using a modified version of the training script of the source provided in Table 16. The pretrained ResNet-18 on ImageNet-21K-P has a Top-1 accuracy of 38.1%, and the model is used for data synthesis and relabel/validation as shown in Table 17 and Table 18, respectively. Note that CutMix used in ImageNet-1K is replaced with CutOut [42], and a relatively large label smooth of 0.2 is used during the ImageNet-21K-P pretraining phase. We incorporate the same changes to the relabel/validation phase of the synthetic dataset.

## C.4 Implementation of Baselines

**RDED.** RDED [8] has several different changes to the SRe$^2$L settings. (1) The batch size is adjusted according to the IPC size (i.e., 100 for IPC10 and 200 for IPC50). (2) It uses additional augmentation (i.e., ShufflePatch to shuffle the position of patches). Such augmentations are considered additional storage since the exact order of patch shuffling needs to be stored. (3) Weaker augmentation (i.e., a larger lower bound for the random area of the resized crop). (4) A smoothed learning rate scheduler. We adhere to all the changes for experiments regarding RDED.

**G-VBSM.** In Table 9, we adopt the several techniques used for G-VBSM [9]. (1) Soft labels are generated with an ensemble of models. Specifically, we use ResNet18 [36], MobileNetV2 [39], EfficientNet-B0 [43], ShuffleNetV2-0.5 [44]. (2) Logit Normalization is used to keep the same label storage. (3) A different MSE+$\gamma \times$GT loss replaces KL divergence, where $\gamma = 0.1$.

# D  Additional Experiments

## D.1  Ablation

Table 19: Ablation study of the proposed method. `C` denotes using class-wise matching. `CS` denotes suing class-wise supervision. `ILP` denotes using an improved label pool. (IPC50, ResNet18, ImageNet-1K).

| +C | +CS | +ILP | 1× | 10× | 20× | 30× | 50× | 100× |
|----|-----|------|-----|-----|-----|-----|-----|------|
| -  | -   | -    | 52.0 | 49.4 | 46.4 | 41.1 | 34.8 | 25.4 |
| ✓  | -   | -    | 54.7 | 51.9 | 48.5 | 42.9 | 37.7 | 22.6 |
| ✓  | -   | ✓    | 54.7 | 52.9 | 49.9 | 46.2 | 40.7 | 27.0 |
| ✓  | ✓   | -    | 55.3 | 53.2 | 49.9 | 45.7 | 39.7 | 29.1 |
| ✓  | ✓   | ✓    | 55.4 | 54.4 | 51.8 | 48.6 | 43.1 | 33.7 |

Table 19 presents an expanded version of Table 4. The row highlighted in grey outlines the ablation study on class-wise supervision, demonstrating that the `ILP` component (Improved Label Pool) enhances performance independently of class-wise supervision.

## D.2  Scaling on Large IPCs

Table 20: Experiment on the scalability of large IPCs. $\mathbb{T}$ denotes the total storage of images and labels, and storage is measured in GB. The validation model is ResNet18.

|  | 1× | $\mathbb{T}$ | 30× | $\mathbb{T}$ | 40× | $\mathbb{T}$ |
|---|-----|-----|-----|-----|-----|-----|
| IPC300 | 65.3 | 178 | 62.6 | 10 | 61.9 | 9 |
| IPC400 | 67.4 | 237 | 65.2 | 13 | 64.6 | 12 |
| ImageNet-1K | 69.8 | 138 | - | - | - | - |

Table 20 demonstrates that our method exhibits commendable scalability across large IPCs. We observe non-marginal enhancements when deploying even larger IPCs, such as IPC300 and IPC400. Moreover, our approach achieves nearly identical accuracy levels — specifically, 65.3% vs. 65.2% — when comparing the use of IPC300 at 1× with IPC400 at 30× less labels. Compared to the full ImageNet-1K dataset, our method preserves a large portion of the accuracy with 10× less storage. This performance is achieved despite the vastly different storage requirements of 178G and 13G, respectively, indicating a higher flexibility of IPC choice with a fixed storage budget.

## D.3  Comparison with Fast Knowledge Distillation [25]

The label quantization technique mentioned in Fast Knowledge Distillation (FKD) [25] is orthogonal to the proposed method for several reasons. Firstly, as demonstrated in Table 21, there are six components related to soft labels. FKD only compresses the prediction logits (component 6), while the our method addresses all six components.

Secondly, even for the overlapping storage component (component 6: prediction logits), the compression targets differ between FKD and our method, as shown in Table 22. The total stored prediction logits can be approximated by the formula: number_of_condensed_images × number_of_augmentations × dimension_of_logits. FKD's label quantization focuses on compressing the dimension_of_logits, whereas the proposed label pruning method focuses on compressing the number_of_augmentations.

Table 21: Different storage components between FKD and the proposed method. FKD, originally for model distillation, requires storage only for components 1, 2, and 6. Adapting it to dataset distillation requires additional storage for components 3, 4, and 5.

| Components of Storage | FKD | Proposed Method |
|---|---|---|
| 1. coordinates of crops | × | ✓ |
| 2. flip status | × | ✓ |
| 3. index of cutmix images | × | ✓ |
| 4. strength of cutmix | × | ✓ |
| 5. coordinates of cutmix bounding box | × | ✓ |
| 6. prediction logits | ✓ | ✓ |

Table 22: Breakdown explanation for component 6 (prediction logits) storage between FKD's label quantization and the proposed label pruning. The number of condensed images is computed by N = IPC × number_of_classes. FKD's compression target is dimension_of_logits, while the proposed method's target is number_of_augmentations.

| Method | Number of Augmentations per Image | Dimension of Logits per Augmentation | Total Storage for Prediction Logits |
|---|---|---|---|
| Baseline (no compression) | 300 | 1,000 | N × 300 × 1000 |
| Label Quantization (FKD) | 300 | 10 | N × 300 × 10 |
| Label Pruning (Proposed) | 3 | 1,000 | N × 3 × 1000 |

Although FKD's approach is orthogonal to our method, a comparative analysis was conducted to better understand their relative performance. Table 23 presents a detailed comparison between FKD's two label quantization strategies (Marginal Smoothing and Marginal Re-Norm) and the proposed method. It is important to note that FKD only compresses component 6, with the compression rate related to hyper-parameter $K$. Components 1-5 remain uncompressed ($1\times$ rate) in FKD. Additionally, FKD's quantized logits store both values and indices, so their actual storage is doubled, and their compression rate is halved.

This analysis has yielded two key observations. First, our method demonstrates higher accuracy at comparable compression rates. For IPC10, our method achieves 32.70% accuracy at $10\times$ compression, while FKD only reaches 18.10% at $8.2\times$ compression. Second, our method exhibits better compression at similar accuracy levels. On IPC10, our method attains 20.20% accuracy at $40\times$ compression, whereas FKD achieves 19.04% at just 4.5x compression.

Table 23: Comparison between FKD's two label quantization strategies (Marginal Smoothing and Marginal Re-Norm) and ours.

| Method | Compression Rate of | | Full Compression Rate | Accuracy (%) on IPC10 |
|---|---|---|---|---|
| | Component 1-5 | Component 6 | | |
| Baseline (no compression) | $1\times$ | $1\times$ | $1\times$ | 34.60 |
| FKD (Smoothing, K=100) | $1\times$ | (10/2)=$5\times$ | $4.5\times$ | 18.70 |
| FKD (Smoothing, K=50) | $1\times$ | (20/2)=$10\times$ | $8.2\times$ | 15.53 |
| FKD (Smoothing, K=10) | $1\times$ | (100/2)=$50\times$ | $23.0\times$ | 9.20 |
| FKD (Re-Norm, K=100) | $1\times$ | (10/2)=$5\times$ | $4.5\times$ | 19.04 |
| FKD (Re-Norm, K=50) | $1\times$ | (20/2)=$10\times$ | $8.2\times$ | 18.10 |
| FKD (Re-Norm, K=10) | $1\times$ | (100/2)=$50\times$ | $23.0\times$ | 15.52 |
| Ours ($10\times$) | $10\times$ | $10\times$ | $10\times$ | **32.70** |
| Ours ($20\times$) | $20\times$ | $20\times$ | $20\times$ | 28.60 |
| Ours ($40\times$) | $40\times$ | $40\times$ | $40\times$ | 20.20 |

# E  Additional Information

## E.1  Label Pruning Metrics

We determine labels according to the statistics of the auxiliary information:

1. `correct`: the number of correctly classified images  [31]
2. `diff`: the absolute difference between the Top-2 outputs
3. `signed_diff`: the signed difference between Top-2 output [45]
4. `cut_ratio`: the cut-mix ratio
5. `confidence`: the value of the largest output [26].

These metrics serve for the baselines compared to random label pruning in Table 5 After knowing the metric, knowing which data type to prune (i.e., "easy", "hard", or "uniform") is important. Additionally, FerKD [26] argues the reliability of generated soft labels and proposes to use neither too easy nor too hard samples.

## E.2  Image and Label Storage

Table 24: Image and label storage. $\mathbb{I}$ denotes image storage. $\mathbb{L}$ denotes label storage. "Ratio" is label-to-image ratio.

| ImageNet-1K (GB) | | | |
|---|---|---|---|
| Storage | $\mathbb{I}$ | $\mathbb{L}$ | Ratio |
| IPC10 | 0.15 | 5.67 | 37.0 |
| IPC20 | 0.30 | 11.33 | 37.6 |
| IPC50 | 0.75 | 28.33 | 37.9 |
| IPC100 | 1.49 | 56.66 | 38.0 |
| IPC200 | 2.98 | 113.33 | 38.0 |
| IPC300 | 4.76 | 172.63 | 36.3 |
| IPC400 | 6.33 | 229.80 | 36.3 |

| Tiny-ImageNet (MB) | | | |
|---|---|---|---|
| Storage | $\mathbb{I}$ | $\mathbb{L}$ | Ratio |
| IPC50 | 21 | 449 | 21.4 |
| IPC100 | 40 | 898 | 22.5 |

| ImageNet-21K-P (GB) | | | |
|---|---|---|---|
| Storage | $\mathbb{I}$ | $\mathbb{L}$ | Ratio |
| IPC10 | 3 | 643 | 214.3 |
| IPC20 | 5 | 1285 | 257.1 |

Table. 24 shows that stored labels are more than $10\times$, $30\times$, and $200\times$ sized of the image storage, depending on the number of classes of the dataset.

## E.3  Theoretical Analysis on the Number of Updates

Our experiments are grounded in a careful analysis of the number of updates required for stable Batch Normalization (BN) statistics. We begin by examining the derived bound from Eq. 8:

$$n \geq \max\left( \underbrace{\frac{-2\ln\left(\frac{T}{2}\right)}{\delta^2 \min(q_c)}}_{\text{Chernoff Bound}}, \quad \underbrace{\frac{\ln\left(\frac{C}{\tau}\right)}{(1-\delta)\varepsilon \min(q_c)}}_{\text{BN Convergence}} \right).$$

To evaluate this bound, we substitute the following values:

- $T = 0.05$ (acceptable total failure probability, corresponding to 95% confidence)
- $\delta = 0.2$ (acceptable relative deviation from the expected number of batches)
- $\varepsilon = 0.1$ (momentum parameter in BN)
- $\min(p_c) = \dfrac{732}{1,281,167} \approx 0.0005711$ (ratio of the least number of images in a class to total images)
- $B = 256$ (batch size)
- $\min(q_c) = 1 - (1 - \min(p_c))^B$ (minimum probability that any class appears in a batch)

First, we compute $\min(q_c)$:

$$
\begin{aligned}
\min(q_c) &= 1 - (1 - \min(p_c))^B \\
&= 1 - (1 - 0.0005711)^{256} \\
&= 1 - (0.9994289)^{256} \\
&\approx 1 - e^{-256 \times 0.0005711} \quad \text{(since } \min(p_c) \text{ is small)} \\
&= 1 - e^{-0.1462} \\
&\approx 1 - 0.8639 = 0.1361.
\end{aligned}
$$

Thus, $\min(q_c) \approx 0.1361$.

Next, we compute the two parts of the bound separately.

**From Chernoff Bound Term:** Given that we allocate the total failure probability $T$ equally between the two events, we have $T/2 = 0.025$.

$$
\begin{aligned}
n &\geq \frac{-2\ln\left(\dfrac{T}{2}\right)}{\delta^2 \min(q_c)} \\
&= \frac{-2\ln(0.025)}{(0.2)^2 \times 0.1361} \\
&= \frac{-2 \times (-3.6889)}{0.04 \times 0.1361} \quad \text{(since } \ln(0.025) = -3.6889) \\
&= \frac{7.3778}{0.005444} \\
&\approx 1,355.2.
\end{aligned}
$$

**From BN Convergence Term:** We need to specify $C$ and $\tau$. Let's assume:

- $C = 1$ (an upper bound on $\left|\text{BN}_c^t - \mu_c\right|$ at initialization, as the running mean is typically initialized to zero)
- $\tau = 0.01$ (desired convergence tolerance)

Compute the numerator:

$$
\ln\left(\frac{C}{\tau}\right) = \ln\left(\frac{1}{0.01}\right) = \ln(100) = 4.6052.
$$

Now, compute the denominator:

$$
(1 - \delta)\varepsilon \min(q_c) = (1 - 0.2) \times 0.1 \times 0.1361 = 0.8 \times 0.1 \times 0.1361 = 0.010888.
$$

Compute the second part:

$$
\begin{aligned}
n &\geq \frac{\ln\left(\dfrac{C}{\tau}\right)}{(1 - \delta)\varepsilon \min(q_c)} \\
&= \frac{4.6052}{0.010888} \\
&\approx 423.08.
\end{aligned}
$$

**Final Bound:**

$$
n \geq \max\left(1,355.2,\ 423.08\right) = 1,355.2 \approx 1,356 \quad \text{(rounding up to the nearest whole number).}
$$

This theoretical result indicates that approximately $1,356$ batches are needed for stable BN statistics with the specified parameters.

**Practical Implications:** This observation leads to a key insight: pretrained models have already undergone sufficient updates to achieve stable BN statistics. Specifically, in the context of ImageNet-1K:

$$\text{Updates per epoch} = \frac{1,281,167}{256} \approx 5,005 \text{ updates} > 1,356.$$

Since one epoch consists of approximately $5,005$ updates, which is substantially more than the theoretical requirement of $1,356$ batches, we can confirm that a single epoch of training is sufficient for the BN statistics of each class to converge within the desired tolerance with high probability.

### E.4    Class-wise Statistics Storage

Table 25: Additional storage required for class-wise statistics. The model is ResNet-18, and storage is measured in MB.

|              | Tiny-ImageNet | ImageNet-1K | ImageNet-21K-P |
|--------------|---------------|-------------|----------------|
| Original     | 43.06         | 44.66       | 247.20         |
| + Class Stats| 50.41         | 81.30       | 445.87         |
| Diff.        | 7.35          | 36.64       | 198.67         |

The additional storage allocation for class-specific statistics is detailed in Table 25. It is observed that this storage requirement escalates with an increase in the number of classes. However, this is a one-time necessity during the recovery phase and becomes redundant once the synthetic data generation is completed.

### E.5    Computing Resources

Experiments are performed on 4 A100 80G GPU cards. We notice that for Tiny-ImageNet, there is a slight performance drop when multiple GPU cards are used with `DataParallel` in PyTorch. Therefore, we use 4 GPU cards for ImageNet-1K and ImageNet-21K-P experiments and 1 GPU card for all Tiny-ImageNet experiments.

### E.6    Limitation and Future Work

We recognize that there are several limitations and potential areas for further investigation. Firstly, while our work significantly reduces the required storage, the process for generating the soft labels is still necessary, as we randomly select from this label space. Secondly, reducing the required labels may not directly lead to a reduced training speed, as the total training epochs remain the same in order to achieve the best performance. Future work is warranted to reduce label storage as well as the required training budget simultaneously.

### E.7    Ethics Statement and Broader Impacts

Our research study focuses on dataset distillation, which aims to preserve data privacy and reduce computing costs by generating small synthetic datasets that have no direct connection to real datasets. However, this approach does not usually generate datasets with the same level of accuracy as the full datasets.

In addition, our work in reducing the size of soft labels and enhancing image diversity can have a positive impact on the field by making large-scale dataset distillation more efficient, thereby reducing storage and computational requirements. This efficiency can facilitate broader access to advanced machine learning techniques, potentially fostering innovation across diverse sectors.

# F  Visualization

In this section, we present visualizations of the datasets used in our experiments. Due to the different matching objectives, datasets of different IPCs should have distinct images. Therefore, we provide the visualization of different IPCs. Figure 6 shows randomly sampled images from ImageNet-1K at various IPC. Figure 7 depicts the Tiny-ImageNet dataset with IPC50 and IPC100. Figure 8 provides visualizations of ImageNet-21K-P at IPC10 and IPC20.

## F.1  ImageNet-1K

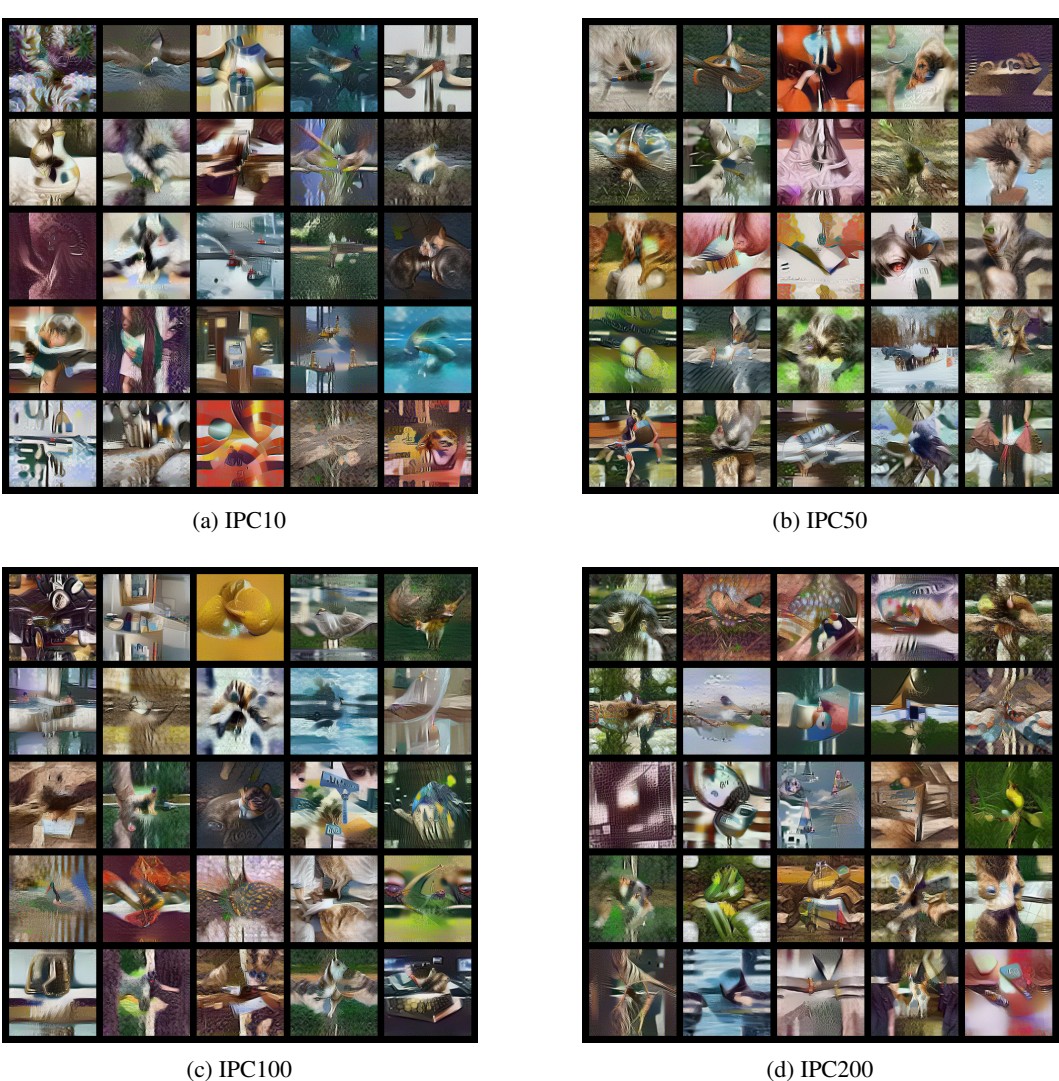

(a) IPC10

(b) IPC50

(c) IPC100

(d) IPC200

Figure 6: Visualization of ImageNet-1K. Images are randomly sampled.

**F.2   Tiny-ImageNet**

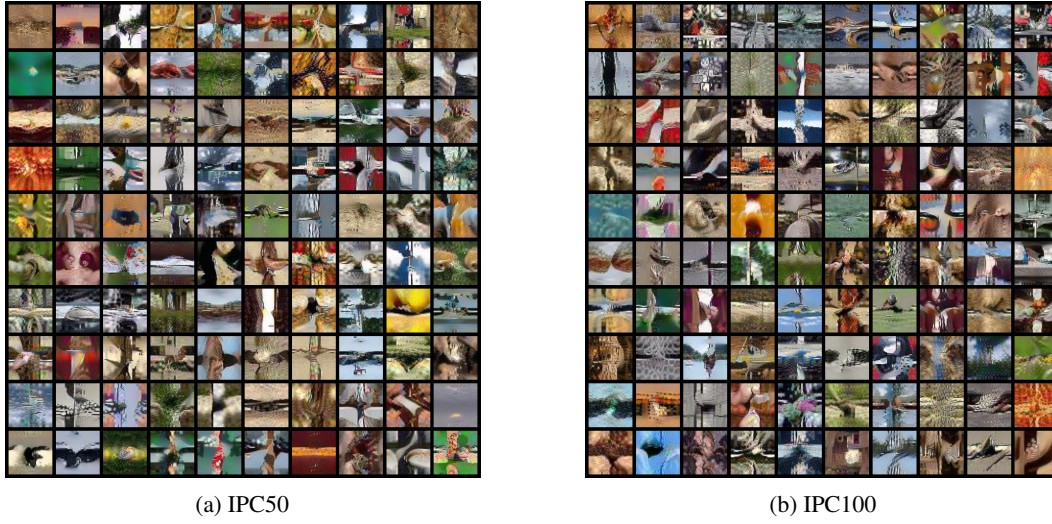

(a) IPC50

(b) IPC100

Figure 7: Visualization of Tiny-ImageNet. Images are randomly sampled.

**F.3   ImageNet-21K-P**

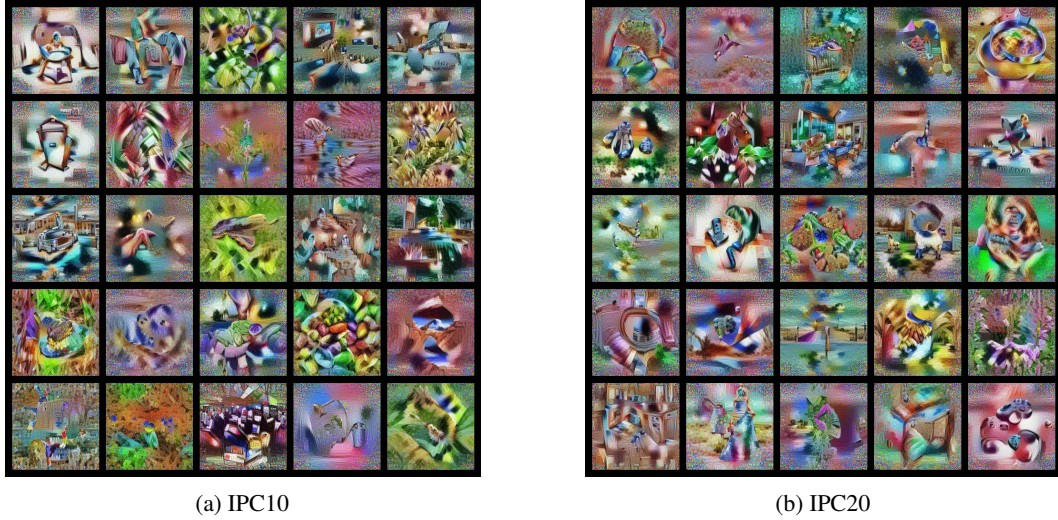

(a) IPC10

(b) IPC20

Figure 8: Visualization of ImageNet-21K-P. Images are randomly sampled.

