# OpenReview forum: "Are Large-scale Soft Labels Necessary for Large-scale Dataset Distillation?"
_NeurIPS.cc/2024/Conference — NeurIPS 2024 poster_

### Official Review · Reviewer_FdYh · 2024-07-05

**Soundness:** 3
**Presentation:** 4
**Contribution:** 3
**Rating:** 7
**Confidence:** 4

**Summary:**

This paper addresses the challenge of large-scale dataset distillation, specifically focusing on reducing the storage requirements for auxiliary soft labels in ImageNet-level condensation. The authors propose Label Pruning for Large-scale Distillation (LPLD), which aims to achieve state-of-the-art performance while significantly reducing the storage needed for soft labels. The main contributions include:
1. Identifying that high within-class similarity in condensed datasets necessitates large-scale soft labels.
2. Introducing class-wise supervision during image synthesis to increase within-class diversity.
3. Demonstrating that simple random pruning of soft labels is effective when data diversity is improved.
4. Achieving SOTA performance with significantly reduced label storage, validated across various networks and datasets.

**Strengths:**

1. The paper addresses an often overlooked problem in dataset distillation: the large storage requirements for auxiliary data. Their class-wise supervision approach offers a novel solution to this issue.

2. The analysis is thorough, using feature cosine similarity and MMD to demonstrate increased diversity in synthetic data. Experiments across Tiny-ImageNet, ImageNet-1K, and ImageNet-21K show the method's effectiveness on various datasets.

3. The paper is well-organized and written. The authors clearly explain their motivations, provide detailed methodology, and offer comprehensive analysis of their results. Their figures and tables effectively illustrate key points.

4. By significantly reducing storage needs (40x compression of soft labels) while maintaining or improving performance, this work makes dataset distillation more practical for large-scale applications.

**Weaknesses:**

Limited theoretical analysis: While the paper provides comprehensive empirical evidence, a more rigorous theoretical analysis could strengthen it. I know this is a common challenge in dataset distillation research. However, the detailed empirical study presented here lays valuable groundwork for potential future theoretical analyses in this direction, particularly regarding the relationship between class-wise supervision and increased diversity.

**Questions:**

1. How sensitive is the method to the choice of hyperparameters, particularly in the label pruning process? Is there a way to automatically determine the optimal pruning rate?

2. Have you explored the potential of combining your class-wise supervision approach with other dataset distillation techniques? Could this lead to further improvements?

3. How does the performance of your method scale with the number of classes? Is there a point where the benefits of class-wise supervision diminish?

4. Have you investigated the impact of your method on the training time of the distilled dataset compared to previous approaches?

5. In Table 6(c), why the performance on Swin-V2-Tiny is the worst even if the architecture has the largest size (28.4M)?

**Limitations:**

Yes. Limitations are provided in Appendix E.5.

---

> ### Author Rebuttal · Authors · 2024-08-06
>
> Thank you for your questions and feedbacks. We want to address them one by one.
>
> > 1. How sensitive is the method to the choice of hyperparameters, particularly in the label pruning process? Is there a way to automatically determine the optimal pruning rate?
>
> Thank you for bringing up this point. Regarding the label pruning process, we adopt two random processes: **(1) random soft label pruning** and **(2) random resampling for training**, as shown in Figure 5. The pruning ratio is the only hyperparameter introduced in this process.
>
> It's important to note that there is no universally optimal pruning rate in this scenario, as the ideal rate depends on the trade-offs between accuracy and storage. From our observations, datasets with higher Images-Per-Class (IPCs) tend to be more robust to label pruning. This indicates that such datasets can better maintain accuracy even when labels are pruned, offering more flexibility in balancing the trade-offs.
>
>
>
> > 2. Have you explored the potential of combining your class-wise supervision approach with other dataset distillation techniques? Could this lead to further improvements?
>
> Thank you for the question. We explored applying our method to CDA, and it shows improvements over the pruning results, making CDA more robust to label pruning, as shown in **Table A**.
>
> **Table A**: Directly applying our method on CDA. ImageNet-1K, IPC50.
>
> |             | 0x   | 10x  | 20x  | 40x  |
> | ----------- | ---- | ---- | ---- | ---- |
> | CDA         | 53.5 | 50.3 | 46.1 | 38.0 |
> | Ours on CDA | 55.2 | 53.4 | 51.0 | 46.0 |
>
>
> Additionally, we believe that initializing with images from RDED [a] could eventually be beneficial. RDED enhances image diversity through an optimization-free approach, and we can further boost image diversity through class-wise supervision.
>
>
> > 3. How does the performance of your method scale with the number of classes? Is there a point where the benefits of class-wise supervision diminish?
>
> Thank you for the question. We do not observe a diminishing performance with an increasing number of classes. **Table B** provides the results of the ImageNet-21K-P dataset (10,450 classes). Despite the increasing challenges of the dataset itself, our method outperforms SRe2L and even CDA by a noticeable margin at 0x baseline. Notably, for IPC20 under a 40x pruning ratio, the performance suffers only a 0.9% drop on ImageNet-21K-P.
>
> **Table B**: Performance of different methods on ImageNet-21K-P. The dataset contains 10,450 classes. This is a simplified version of Table 7 from our paper.
>
> | ImageNet-21K-P | SRe2L (1x) | CDA (1x) | Ours (1x) | Ours (40x) |
> | -------------- | ---------- | -------- | --------- | ---------- |
> | IPC10          | 18.5       | 22.6     | 25.4      | 21.3       |
> | IPC20          | 20.5       | 26.4     | 30.3      | 29.4       |
>
>
>
> > 4. Have you investigated the impact of your method on the training time of the distilled dataset compared to previous approaches?
>
> Thank you for the insightful question; this is worth exploring. Currently, we have not changed the total training epochs (e.g., 300 epochs for ImageNet-1K), so there is no apparent difference in training the distilled dataset. The only difference is that SRe2L uses the full set of labels while we reuse a subset of the labels. The reuse of labels has almost no impact on the training time.
>
>
> > 5. In Table 6(c), why the performance on Swin-V2-Tiny is the worst even if the architecture has the largest size (28.4M)?
>
> Table 6(c) presents the results of cross-architecture performance. Specifically, synthetic data are recovered using ResNet18 and evaluated on other networks. Poor performance on Swin-V2-Tiny may be due to its different network structure (i.e., transformer-based).
>
> 1. Transformer-based networks may adapt less effectively to CNNs, especially since all the information is obtained from ResNet.
> 2. Additionally, transformer-based networks are well-known for being data-hungry, and distilled datasets contain even less data. Therefore, Swin-V2-Tiny may exhibit the poorest performance despite having the largest model size.
>
> We note that a similar pattern is observed in RDED [a], as shown in **Table C**.
>
> **Table C**: Cross-architecture performance. Data is obtained from Table 5 of RDED [a].
>
> | Recover: ResNet-18 | Model Size | SRe2L      | RDED       |
> | ------------------ | ---------- | ---------- | ---------- |
> | ResNet-18          | 11.7M      | 21.7 ± 0.6 | 42.3 ± 0.6 |
> | EfficientNet-B0    | 5.3M       | 25.2 ± 0.2 | 42.8 ± 0.5 |
> | MobileNet-V2       | 3.5M       | 19.7 ± 0.1 | 34.4 ± 0.2 |
> | Swin-V2-Tiny       | 28.4M      | 9.6 ± 0.3  | 17.8 ± 0.1 |
>
>
> I hope the responses address your concerns. Thank you!
>
> [a] Sun, Peng, et al. "On the diversity and realism of distilled dataset: An efficient dataset distillation paradigm." *CVPR, 2024.*

---

> > ### Comment · Reviewer_FdYh · 2024-08-10
> > **After rebuttal**
> >
> > Thanks for providing the detailed rebuttal. All of my concerns have been properly addressed. In addition, I have also read the other reviews’ comments and corresponding rebuttals. These efforts are appreciated.
> >
> > After careful consideration, I believe this is a high-quality paper that merits acceptance. The proposed soft label compression technique is valuable and contributes significantly to the field.
> >
> > I will adjust my score accordingly once the discussions between other reviewers and the authors have concluded.

---

> > > ### Author Response · Authors · 2024-08-13
> > > **Thank you for your positive feedback**
> > >
> > > Thank you sincerely for your positive feedback on our rebuttal. We find it very encouraging that our paper is considered high-quality and that our soft label compression technique is recognized as a significant contribution to the field.
> > >
> > > As we are approaching the discussion deadline, we wanted to provide a brief update: one reviewer has already raised their score after reading the rebuttal, while another has not yet responded. Given your intention to adjust the score and your positive assessment, we were wondering if you might consider updating your score at this time. Your support could be crucial in the final evaluation of our paper.
> > >
> > > We deeply appreciate your time and expertise throughout this review process. Thank you again for your thoughtful consideration.

---

> > > > ### Comment · Reviewer_FdYh · 2024-08-13
> > > >
> > > > Thanks for your follow-up. After careful consideration, I have decided to raise my score and strongly support the acceptance of this paper. I believe this work is significant to the field.

---

> > > > > ### Author Response · Authors · 2024-08-13
> > > > >
> > > > > Thank you for your strong support. We greatly appreciate your recognition of our work's significance to the field. Your opinion is extremely important to us, and we are deeply grateful.

---

### Official Review · Reviewer_8MXG · 2024-07-08

**Soundness:** 2
**Presentation:** 3
**Contribution:** 2
**Rating:** 5
**Confidence:** 4

**Summary:**

The paper focuses on reducing the size of soft-label storage in large-scale dataset condensation. The authors discussed why the labels are large-scaled and then proposed to prune the labels by increasing the diversity of synthetic images. Extensive experiments are conducted to validate the effectiveness of the proposed method.

**Strengths:**

1. The paper tackles the problem of large storage space consumed by soft-labels, and discover that this is attributed to the lack of image diversity within classes.
2. Extensive experiments demonstrate the effectiveness of the proposed method, achieving comparable or even better performance than previous methods with reduced label storage.

**Weaknesses:**

The motivation of the work is well-grounded. I don’t observe major drawbacks but do have some minor concerns towards its technical contributions.

1. The proposed method seems to be a class-wise version of SRe^2L, with a soft label-reuse mechanism during training. Thus the technical contributions seem to be weak.
2. How is Equation (8) used in the proposed method? Or is it just a theoretical result?
3. I have doubts on the computational cost (memory and time) of the proposed method (not storage), doing class-wise optimization may increase the distillation time, especially on ImageNet.
4. How is the random label pruning done? E.g. Is the same number of soft-labels removed for each epoch?
5. Proposition 2 is quite confusing, is “higher MMD” a typo?

**Questions:**

Please see Weaknesses.

**Limitations:**

Yes, the limitations are addressed.

---

> ### Author Rebuttal · Authors · 2024-08-06
>
> Thank you for bring up all these values feedbacks.
>
> > 1. The proposed method seems to be a class-wise version of SRe^2L, with a soft label-reuse mechanism during training. Thus the technical contributions seem to be weak.
>
> Thank you for your important question.
> Our approach is not merely a variant of existing methods but a comprehensive redesign that addresses key limitations. We'd like to elaborate on our key contributions and the novelty of our work:
>
> 1. **Novel Motivation and Insight**. We recognized the critical importance of image diversity in large-scale distillation tasks. This insight led us to rethink the fundamental approach to distillation tasks.
> 2. **Unified Framework for Diversity and Pruning**. We uniquely combine image diversity enhancement and label pruning synergistically.
> 3. **Simplicity**. The strength of our approach lies in its simplicity. We demonstrate that **random** **pruning** within our class-wise framework is sufficient, eliminating the need for complex pruning metrics.
> 4. **Effectiveness**.  Our method enhances performance with full labels. Additionally, our method can maintain comparable results at a 40x compression ratio, demonstrating its robustness across various data scenarios. This shows that rethinking the problem from a class-wise angle can lead to significant improvements.
>
> > 2. How is Equation (8) used in the proposed method? Or is it just a theoretical result?
>
> Thanks for your thoughtful question. Our experiments are grounded in a careful analysis of the number of updates required for stable BN statistics.
>
> First, for Equation 8:
>
> $$n \geq \max \left(\frac{-2 \ln (T)}{\delta^2 \min \left(p_c\right)}, \frac{\ln (T)}{(1-\delta) \ln (1-\varepsilon) \cdot \min \left(p_c\right)}\right)$$
>
> Let’s substitute the values and compute the two parts one by one,
>
> 1. $T=0.05$ (95% confidence)
> 2. $\delta=0.2$ (moderate sigma to handling class imbalance)
> 3. $\epsilon=0.1$ (default value in BN)
> 4. $p_c=(732/1,281,167)= 0.0005711$  = least number of images in a class / total images
>
> $$n \geq \max \left( \frac{-2 \times \ln(0.05)}{0.2^2 \times 0.0005711},  \frac{\ln(0.05)}{(1 - 0.2) \ln(1 - 0.1) \cdot 0.0005711}\right) = \max (262234, 62234) = 262,234$$
>
> Next, let us elaborate on how we use the theoretical result to guide the experiment design:
>
> 1. $n \geq 262,234$ means that the theoretical number of updates needed for stable BN statistics is $262,234$.
> 2. In ResNet training on ImageNet-1K, the standard setting [a] uses a batch size of 256 and trains for 90 epochs. The total number of updates is $(1,281,167/256) \times 90 = 450,411$. This significantly exceeds our theoretical requirement of 262,234 updates.
> 3. This observation gave us a key insight: pretrained models have already undergone sufficient updates to achieve stable BN statistics. Therefore, we can recompute class-wise BN statistics using a pretrained model for only **one epoch** (5,005 updates).
>
> [a] He, Kaiming, et al. "Deep residual learning for image recognition." *CVPR*. 2016.
>
> > 3. I have doubts on the computational cost (memory and time) of the proposed method (not storage), doing class-wise optimization may increase the distillation time, especially on ImageNet.
>
> Thank you for bringing up your concern. We acknowledge that our method can indeed increase computational costs in some cases. However, the impact varies depending on the Images-Per-Class (IPC) setting. Based on our experiments in **Table A**, we have:
> - **IPC 50**:  19.95% slower than SRe2L.
> - **IPC 100**: 7.28% slower than SRe2L.
> - **IPC 200**: 4.83% **faster** than SRe2L.
>
> On average, the computational time increase is 7.47%. This suggests that while there is indeed a computational cost increase in some scenarios, the impact is not uniform and our method can even be more efficient in certain configurations.
>
> We believe this trade-off is exceptionally favorable: a minimal **7.47%** increase in processing time unlocks a dramatic **40x** reduction in data volume. Such powerful compression capability far outweighs the marginal computational cost, making our approach particularly valuable for large-scale applications where data efficiency is paramount.
>
> **Table** **A**: Real computation cost of SRe2L and our method on ImageNet-1K. The time spent on inner loops is averaged from 3 loops.  The optimization process contains two loops: the outer loop and the inner loop.  Total time = (Time for Inner Loop * Outer Loop Numbers).
>
> |  | IPC 50 | IPC 100 | IPC 200 |
> |---|---|---|---|
> | SRe2L - Breakdown | 37m5s $\times$ 50 | 37m5s $\times$ 100 | 37m5s $\times$ 200 |
> | SRe2L - Total | 30h55m | 61h49m | 123h37m |
> | Ours - Breakdown | 2m13s$\times$ 100 | 3m59s $\times$ 1000 | 7m4s $\times$ 1000 |
> | Ours - Total | 37h5m | 66h19m | 117h39m |
>
> > 4. How is the random label pruning done? E.g. Is the same number of soft-labels removed for each epoch?
>
> Thank you for this important question.
> Yes,  the number of soft labels removed for each epoch is the **same**. The detailed process for random label pruning is shown in Figure 5. There are two different levels for random pruning:
>
> 1. First-level random pruning for the soft label pool: **9 samples** from **12 samples**.
> 2. Second-level random pruning for training: **3 samples** from **9 samples**.
>
> For first-level random pruning, we introduce two different schemes:
>
> 1. Epoch-level random pruning: if 1st sample is removed, 2nd 3rd must be removed.
> 2. Batch-level random pruning: even if 1st is removed, 2nd 3rd can be randomly removed or kept.
>
> In our paper, for first-level random pruning, we stick to batch-level random pruning as it introduces more randomness. For second-level random pruning, we use normal randomness.
>
>
> > 5. Proposition 2 is quite confusing, is “higher MMD” a typo?
>
> Thank you so much for bringing out this typo. We have corrected this error in the original manuscript. Also, we have conducted a thorough review of the entire document to ensure no similar errors exist.

---

> > ### Comment · Reviewer_8MXG · 2024-08-10
> > **Thank you for the response**
> >
> > Thank you for the detailed response and my concerns are addressed. I now vote for acceptance of this paper. However, I strongly recommend the authors to include the discussions on time cost and detailed pruning process into the main paper/appendix. These will be insightful for the community.

---

> > > ### Author Response · Authors · 2024-08-11
> > > **Thank you for the acceptance**
> > >
> > > We are deeply grateful for your acceptance of our paper and truly appreciate your insightful feedback. Your comment that our discussion "will be insightful for the community" is particularly encouraging to us.
> > >
> > > We are committed to improving our manuscript based on your thoughtful suggestions.

---

### Official Review · Reviewer_S5vb · 2024-07-12

**Soundness:** 2
**Presentation:** 2
**Contribution:** 2
**Rating:** 5
**Confidence:** 3

**Summary:**

This paper discovers that the conventional method generates images with high similarity. To solve this, the authors introduce class-wise supervision during the image-synthesizing process by batching the samples within classes. Thanks to the increase in diversity, the soft labels can be pruned to reduce the storage size. Extensive experiments are performed on ImageNet with different compression ratios of labels to validate the effectiveness of the method.

**Strengths:**

1. Improving the diversity in condensed images is crucial for obtaining high performance.

2. Label pruning is necessary to reduce the storage size for soft labels.

**Weaknesses:**

1. There are confusing sentences. The authors mention that ”The high similarity of images within the same class requires extensive data augmentation to provide different supervision” and then “To address this issue, we propose Label Pruning for Large-scale Distillation”. How does label pruning address the issue? I see that increasing the diversity of synthetic samples does improve the performance and label pruning is proposed to reduce the storage size. These are not highly related.
2. The authors should compare the performance of the proposed Label pruning with other compression methods (e.g., Marginal Smoothing/Re-Norm with Top-K, using different K when targeting at 10x/20x…) proposed in FKD. Without comparing these methods, it is unclear whether the proposed approach is better than previous pruning methods.


FKD: A Fast Knowledge Distillation Framework for Visual Recognition, Zhiqiang Shen et al

**Questions:**

See weaknesses

---

> ### Author Rebuttal · Authors · 2024-08-07
>
> Thank you so much for raising these questions and concerns. We want to address them one by one.
>
> > 1. There are confusing sentences. The authors mention that ”The high similarity of images within the same class requires extensive data augmentation to provide different supervision” and then “To address this issue, we propose Label Pruning for Large-scale Distillation”. How does label pruning address the issue? I see that increasing the diversity of synthetic samples does improve the performance and label pruning is proposed to reduce the storage size. These are not highly related.
>
> Thank you for your insightful question.
>
> ```
> Previous:      High Similarity -> More Augmentation -> More Label with
>                                                        High Similarity
>
> Naive Pruning: High Similarity -> More Augmentation -> More Label with ------> Fewer Label with
>                                                        High Similarity         High Similarity
>
> Ours LPLD:     Low Similarity  -> Less Augmentation -> Fewer Label with
>                                                        Low Similarity
> ```
>
> We draw the above diagram to clarify the relationship between image similarity and our proposed Label Pruning for Large-scale Distillation (LPLD) method.
>
> First, it's important to distinguish between naive pruning and our LPLD pruning:
>
> 1. **Naive pruning**: This method directly prunes soft labels generated by previous approaches like SRe2L and CDA, starting with **high-similarity** images. It does not address the issue of image diversity.
> 2. **Our LPLD** pruning: We begin with **low-similarity** images and then apply label pruning. This two-step process addresses both image diversity and storage efficiency.
>
> When the reviewer mentions the lack of **direct relation** between pruning and image similarity, we believe this observation is most applicable to naive pruning methods, not our LPLD. Our approach, however, takes a fundamentally different direction:
>
> 1. We first enhance image diversity by creating **low-similarity** images. This step creates a prerequisite for effective **pruning**.
> 2. We then apply label pruning to this diverse set of images, which allows for more effective compression while maintaining performance.
>
> To illustrate the effectiveness of **connecting image similarity and pruning**, we conducted a comparison between naive pruning (**NOT related** to similarity) and our LPLD (**related** to similarity) for ImageNet-1K at IPC50, as shown in **Table A**. The results demonstrate that our method outperforms naive pruning, especially at higher compression rates.
>
> **Table A**: Comparison between naive pruning on previous methods and our LPLD pruning for ImageNet-1K at IPC50.
>
> | Compression Rate | 0x | 10x | 20x | 30x | 40x |
> |---|---|---|---|---|---|
> | SRe2L | 41.10% | 40.30% | 39.00% | 34.60% | 29.80% |
> | CDA | 48.70% | 45.00% | 41.20% | 35.80% | 30.90% |
> | Ours | **48.80%** | **46.70%** | **44.30%** | **40.20**% | **38.40%** |
>
> In summary, our LPLD method addresses the issue of high image similarity within classes by creating diverse images first and then applying label pruning. This approach improves both the diversity of synthetic samples and reduces storage size, addressing both concerns at the same time.
>
> > 2. The authors should compare the performance of the proposed Label pruning with other compression methods (e.g., Marginal Smoothing/Re-Norm with Top-K, using different K when targeting at 10x/20x…) proposed in FKD. Without comparing these methods, it is unclear whether the proposed approach is better than previous pruning methods.
>
> Thank you for your question.
> Although FKD's approach is orthogonal to our method (reasons for the orthogonality are listed in another response comment), we compared different target components in **Table B** and conducted a comparative analysis presented in **Table C**. Table C follows the definition of the components in Table B. Please note that FKD only compresses component 6, with the compression rate related to hyper-parameter $K$. Components 1-5 remain uncompressed (1x rate). We achieve better performance:
>
> 1. Higher Accuracy at Comparable Compression Rates: For IPC10, our method achieves 32.70% accuracy at 10x compression, while FKD only reaches 18.10% at 8.2x compression.
>
> 2. Better Compression at Similar Accuracy Levels: On IPC10, our method attains 20.20% accuracy at 40x compression, whereas FKD achieves 19.04% at just 4.5x compression.
>
> We hope this response addresses your concern.
>
> **Table B**: Different target components between FKD and ours. FKD, originally for **model** distillation,  requires storage only for components 1, 2, 6. Adapting it to **dataset** distillation requires additional storage for components 3, 4, 5.
> |  | FKD | Ours |
> |---|---|---|
> | 1. coordinates of crops | ❌ | ✔ |
> | 2. flip status | ❌ | ✔ |
> | 3. index of cutmix images | ❌ | ✔ |
> | 4. strength of cutmix | ❌ | ✔ |
> | 5. coordinates of cutmix bounding box | ❌ | ✔ |
> | 6. prediction logits | ✔ | ✔ |
>
> **Table C**: Comparison between FKD's two label quantization strategies (Marginal Smoothing and Marginal Re-Norm) and ours.  FKD’s quantized logits store both values and indices, so their actual storage is doubled, and their compression rate is halved.
>
> | Method | Compression rate of component 1-5 | Compression rate of component 6 | Full compression rate | Accuracy (%) on IPC10 |
> |---|---|---|---|---|
> | Baseline (no compression) | 1x | 1x | **1x** | 34.60 |
> | FKD (Smoothing, K=100) | 1x | (10/2)=5x | **4.5x** | 18.70 |
> | FKD (Smoothing, K=50) | 1x | (20/2)=10x | **8.2x** | 15.53 |
> | FKD (Smoothing, K=10) | 1x | (100/2)=50x | **23.0x** | 9.20 |
> | FKD (Re-Norm, K=100) | 1x | (10/2)=5x | **4.5x** |  19.04 |
> | FKD (Re-Norm, K=50) | 1x | (20/2)=10x | **8.2x** | 18.10 |
> | FKD (Re-Norm, K=10) | 1x | (100/2)=50x | **23.0x** | 15.52 |
> | Ours (10x) | 10x | 10x | **10x** | **32.70** |
> | Ours (20x) | 20x | 20x | **20x** | 28.60 |
> | Ours (40x) | 40x | 40x | **40x** | 20.20 |

---

> ### Author Response · Authors · 2024-08-07
> **The reasons why label quantization in FKD is orthogonal to our method**
>
> Thank you for bringing up the questions related to FKD.
>
> We would like to emphasize that label quantization mentioned in FKD is orthogonal to our method for the following reasons:
>
> 1. **We consider more** **components** **as shown in Table B.** There are **six** **components** related to soft labels. FKD only compresses the prediction logits (component 6), while our method addresses all six components.
> 2. **We have different compression targets, as shown in Table D**: Even for the overlapping storage component (component 6: prediction logits), our compression target differs from FKD's. The total stored prediction logits can be approximated by the `number_of_condensed_images × number_of_augmentations × dimension_of_logits`.
>
> - FKD's **Label Quantization** focuses on compressing the `dimension_of_logits`.
> - Our **Label Pruning** method focuses on compressing the `number_of_augmentations`.
>
> **Table** **B**: Different storage components between FKD and ours. FKD, originally for **model** distillation, requires storage only for components 1, 2, 6. Adapting it to **dataset** distillation requires additional storage for components 3, 4, 5.
> |  | FKD | Ours |
> |---|---|---|
> | 1. coordinates of crops | ❌ | ✔ |
> | 2. flip status | ❌ | ✔ |
> | 3. index of cutmix images | ❌ | ✔ |
> | 4. strength of cutmix | ❌ | ✔ |
> | 5. coordinates of cutmix bounding box | ❌ | ✔ |
> | 6. prediction logits | ✔ | ✔ |
>
> **Table D:** Breakdown explanation for component 6 (prediction logits) storage between FKD’s label quantization and our label pruning. The number of condensed images is computed by `N = IPC x number_of_classes`. FKD's compression target is `dimension_of_logits`, while our target is `number_of_augmentations`.
>
> |                           | Number of condensed images | Number  of augmentations per image | Dimension of logits per augmentation | Total storage for prediction logits |
> | ------------------------- | ------------------------ | ------------------------------- | ------------------------------- | ----------------------------------- |
> | Baseline (no compression) | N                        | 300                             | 1,000                           | N $\times$ 300 $\times$ 1000        |
> | Label Quantization (FKD)  | N                        | 300                             | 10                              | N $\times$ 300 $\times$ 10          |
> | Label Pruning (Ours)      | N                        | 3                               | 1,000                           | N $\times$ 3 $\times$ 1000          |

---

> > ### Author Response · Authors · 2024-08-12
> > **Grateful for any feedback**
> >
> > Dear Reviewer S5vb,
> >
> > We greatly appreciate the time and effort you've invested in reviewing our work. As we are now two days away from the discussion deadline, we wanted to reach out regarding our rebuttal, which we submitted earlier.
> >
> > If possible, we would be grateful for any feedback you might be able to provide, as this would give us the opportunity to engage in a productive discussion before the deadline. We're eager to address any remaining questions or concerns you may have about our paper.
> >
> > We understand that you likely have a busy schedule, and we truly appreciate your valuable insights and expertise in this process.
> >
> > Thank you for your time and consideration.

---

> > > ### Author Response · Authors · 2024-08-13
> > > **Grateful for any feedback**
> > >
> > > Dear Reviewer S5vb,
> > >
> > > I apologize for reaching out again after my previous message. With the discussion deadline now just one day away, I wanted to very gently follow up on our earlier correspondence regarding our paper and rebuttal.
> > >
> > > We completely understand if you haven't had the opportunity to review our rebuttal yet, given the many demands on your time. However, if you have had the chance to look it over, we would be immensely grateful for any feedback you could provide. Your insights, no matter how concise, would be valuable in helping us engage in a constructive discussion before the deadline closes.
> > >
> > > We are sincerely thankful for your continued involvement in this process, and we respect your time and professional commitments. Thank you for your understanding and consideration.

---

> > > > ### Comment · Reviewer_S5vb · 2024-08-13
> > > >
> > > > Thanks for providing additional experiments using label quantization proposed in FKD. Most of my concerns have been addressed. Thus, I increase the score.

---

> > > > > ### Author Response · Authors · 2024-08-13
> > > > > **Thank you for raising the score**
> > > > >
> > > > > Thank you for your prompt response and for reviewing our additional experiments. We're delighted that our efforts have addressed most of your concerns and appreciate your decision to increase the score.
> > > > >
> > > > > Your feedback has been invaluable in improving our work. If you have any further questions, please don't hesitate to ask.
> > > > > Thank you again for your time and expertise.

---

### Decision · Program_Chairs · 2024-09-25

**Decision:**

Accept (poster)

**Comment:**

This paper received all positive recommendations.  Reviewers acknowledged the motivation of this paper,  the thorough analysis, and the significant results. Most questions were well addressed during the rebuttal and discussion. The AC agrees with the acceptance recommendation.